# Going Beyond Linear Transformers
# with Recurrent Fast Weight Programmers

**Kazuki Irie**[1]*, **Imanol Schlag**[1]*, **Róbert Csordás**[1], **Jürgen Schmidhuber**[1,2]
[1]The Swiss AI Lab, IDSIA, University of Lugano (USI) & SUPSI, Lugano, Switzerland
[2]King Abdullah University of Science and Technology (KAUST), Thuwal, Saudi Arabia
{kazuki, imanol, robert, juergen}@idsia.ch

## Abstract

Transformers with linearised attention ("linear Transformers") have demonstrated the practical scalability and effectiveness of outer product-based Fast Weight Programmers (FWPs) from the '90s. However, the original FWP formulation is more general than the one of linear Transformers: a *slow* neural network (NN) continually reprograms the weights of a *fast* NN with *arbitrary* architecture. In existing linear Transformers, both NNs are feedforward and consist of a single layer. Here we explore new variations by adding recurrence to the slow and fast nets. We evaluate our novel recurrent FWPs (RFWPs) on two synthetic algorithmic tasks (code execution and sequential ListOps), Wikitext-103 language models, and on the Atari 2600 2D game environment. Our models exhibit properties of Transformers and RNNs. In the reinforcement learning setting, we report large improvements over LSTM in several Atari games. Our code is public.[1]

## 1 Introduction

The Transformer [1] has become one of the most popular neural networks (NNs) for processing sequential data. Its success on neural machine translation quickly transferred to other problems in natural language processing, such as language modelling [2, 3] or question answering [4]. Recently, it has also been applied in other domains, including image processing [5, 6] or mathematical problem solving [7, 8, 9].

Conceptually, the Transformer is a deep feedforward NN that processes all elements of a sequence in parallel: unlike in recurrent NNs (RNNs), the computations of a layer for the entire sequence can be packed into one big matrix multiplication. This scales well with the number of parallel processors.

Despite the benefits of parallelisation, a major drawback of Transformers is that their computational complexity in time and space is quadratic in sequence length. Furthermore, in the auto-regressive version [1, 2] — the focus of our work — the state size increases linearly with sequence length. This makes Transformers infeasible for auto-regressive settings dealing with very long or potentially infinite sequences, forcing practitioners to truncate temporal contexts and ignore long-term dependencies beyond fixed-size time windows. Although recent work tries to address this issue [10, 11], this limitation makes some applications of Transformers challenging, e.g., reinforcement learning (RL) in partially observable environments [12, 13], which is still dominated by RNNs such as the Long Short-Term Memory (LSTM; [14]) trained by policy gradients [15, 16, 17, 18].

To scale Transformers to longer sequences, recent works have proposed to linearise the softmax in the self-attention computation and reorganise the latter in a sequential way [19]. Such models include

---

*Equal contribution.
[1]https://github.com/IDSIA/recurrent-fwp

35th Conference on Neural Information Processing Systems (NeurIPS 2021).

Katharopoulos et al.'s *Linear Transformer* (LT) [19], Choromanski et al.'s *Performer* [20] and Peng et al. [21]'s variant. They enjoy time and space complexities linear in sequence length with states of constant size. While their performance on some tasks does not fully match the one of regular Transformers [22], several improvements have already been proposed [21, 23] (see our review in Sec. 2.2) which makes this Transformer family a promising alternative.

Here we go one step further in advancing linear Transformer variants as powerful auto-regressive sequence processing models, adopting the perspective of "Fast Weight Programmers" (FWPs) [24, 25, 26]. Recent work emphasised that linearised Transformers are essentially equivalent to outer product-based FWPs from the '90s ([23]; reviewed in Sec. 2). Here we explore this connection further and describe more powerful FWPs.

The original FWP [24] is a two-NN system: a slow and a fast net, each with arbitrary architectures. The slow net learns to generate rapid context-dependent weight modifications for the fast net. In the case of existing linear Transformer variants, the slow and fast nets are simple one layer feedforward NNs. Here we augment them with recurrent connections to obtain recurrent FWPs (RFWPs). Recurrence enhances the model's theoretical power [27] and can help to solve tasks that naturally require recurrence as a part of the solution.

Our experiments on the language modelling dataset Wikitext-103 [28] show that our RFWPs are competitive compared to regular Transformers. We then study various properties of the proposed models on two synthetic algorithmic tasks: code execution [29] and sequential ListOps [30]. Finally, it is straightforward to apply our models to RL problems as a drop-in replacement for LSTMs. Here our RFWPs obtain large improvements over LSTM baselines across many Atari 2600 2D game environments [31]. Although LSTM still works better in a few environments, we show that our RFWPs generally improve by scaling them up.

The main contribution of this work is twofold: (1) from the perspective of FWPs, we study novel powerful FWPs for sequence processing, demonstrating that NNs can easily learn to control NNs that are more complex than a single feedforward layer, and (2) from the perspective of Transformer models, our RFWPs augment linear Transformers with recurrence, addressing general limitations of existing auto-regressive Transformer models.

## 2 Background on Fast Weight Programmers (FWPs)

Here we review the general concept of FWPs, as well as two specific instances thereof: the linear Transformer [19, 20] and the Delta Net [23].

### 2.1 General Formulation

We refresh the concept of fast weight controllers or FWPs [24, 25] using modern notation in a sequence processing scenario. An FWP with trainable parameters $\boldsymbol{\theta}_{\text{slow}}$ sequentially transforms an input sequence $\{\boldsymbol{x}^{(t)}\}_{t=1}^{T}$ with $\boldsymbol{x}^{(t)} \in \mathbb{R}^{d_{\text{in}}}$ to an output sequence $\{\boldsymbol{y}^{(t)}\}_{t=1}^{T}$ with $\boldsymbol{y}^{(t)} \in \mathbb{R}^{d_{\text{out}}}$ of length $T$ as

$$\boldsymbol{\theta}_{\text{fast}}^{(t)}, \boldsymbol{q}^{(t)} = \texttt{SlowNet}\left(\{\boldsymbol{x}^{(j)}\}_{j=1}^{t}, \{\boldsymbol{y}^{(j)}\}_{j=0}^{t-1}, \{\boldsymbol{\theta}_{\text{fast}}^{(j)}\}_{j=0}^{t-1}, \{\boldsymbol{q}^{(j)}\}_{j=0}^{t-1}; \boldsymbol{\theta}_{\text{slow}}\right) \quad (1)$$

$$\boldsymbol{y}^{(t)} = \texttt{FastNet}\left(\{\boldsymbol{q}^{(j)}\}_{j=1}^{t}, \{\boldsymbol{y}^{(j)}\}_{j=0}^{t-1}; \boldsymbol{\theta}_{\text{fast}}^{(t)}\right) \quad (2)$$

where $\boldsymbol{y}^{(0)}$, $\boldsymbol{\theta}_{\text{fast}}^{(0)}$, and $\boldsymbol{q}^{(0)}$ are initial variables. This is a system with two NNs called `FastNet` and `SlowNet` in which the parameters $\boldsymbol{\theta}_{\text{fast}}^{(t)}$ of `FastNet` are generated by `SlowNet` at each time step $t$. The weights of the fast net are *fast* in the sense that they may rapidly change at every step of the sequence while the weights of the slow net $\boldsymbol{\theta}_{\text{slow}}$ are *slow* because they can only change through gradient descent during training, remaining fixed afterwards[2]. Eq. 1 expresses a slow NN in its general form. The slow net can generate fast weights conditioned on various variables, depending on architectural choices for the slow and fast NNs. In addition to the fast weights $\boldsymbol{\theta}_{\text{fast}}^{(t)}$, the slow net also generates or *invents* an input $\boldsymbol{q}^{(t)}$ to be fed to the fast net (alternatively $\boldsymbol{q}^{(t)}$ can simply be $\boldsymbol{x}^{(t)}$). While the architectures of slow and fast nets are arbitrary, they are typically chosen to be differentiable such that the entire FWP can be trained in an end-to-end manner using gradient descent. By interpreting the weights of

---

[2]The fast net could also contain some additional slow weights; we omit this possibility here.

an NN as a program [32], the slow net effectively learns to control, or *program*, the fast NN. Thus, the slow net is a neural programmer of fast weights, and its parameter set $\boldsymbol{\theta}_{\text{slow}}$ embodies compressed information used to produce potentially infinite variations of context-dependent fast weights.

In many settings, it makes sense to generate the fast weights $\boldsymbol{\theta}_{\text{fast}}^{(t)}$ incrementally in an iterative fashion, where the `SlowNet` is further decomposed into two sub-parts:

$$\boldsymbol{z}^{(t)}, \boldsymbol{q}^{(t)} = \texttt{SlowSubnet}(\{\boldsymbol{x}^{(j)}\}_{j=1}^{t}, \{\boldsymbol{y}^{(j)}\}_{j=0}^{t-1}, \{\boldsymbol{\theta}_{\text{fast}}^{(j)}\}_{j=0}^{t-1}, \{\boldsymbol{q}^{(j)}\}_{j=0}^{t-1}, \{\boldsymbol{z}^{(j)}\}_{j=0}^{t-1}; \boldsymbol{\theta}_{\text{slow}}) \qquad (3)$$

$$\boldsymbol{\theta}_{\text{fast}}^{(t)} = \texttt{UpdateRule}(\boldsymbol{\theta}_{\text{fast}}^{(t-1)}, \boldsymbol{z}^{(t)}) \qquad (4)$$

where `UpdateRule` takes the fast weights $\boldsymbol{\theta}_{\text{fast}}^{(t-1)}$ from the previous iteration to produce the new fast weights $\boldsymbol{\theta}_{\text{fast}}^{(t)}$ conditioned on $\boldsymbol{z}^{(t)}$. The update rule is essentially the differentiable *elementary programming instruction* used by the FWP. In the next section we review concrete examples of recent FWPs.

## 2.2 Linear Transformers as Fast Weight Programmers

In general, the dimension of the fast weights $\boldsymbol{\theta}_{\text{fast}}^{(t)}$ is too large to be conveniently parameterised by an NN. Instead, it was proposed in 1991 [24] to perform a rank-one update via the outer product of two vectors generated by the slow net. Two recent models directly correspond to such outer product-based FWPs: linear Transformers [19] and the Delta Net [23].

**Linear Transformer.** The "linear Transformer" [19] is a class of Transformers where the softmax in the attention is linearised. This is achieved by replacing the softmax with a kernel function $\phi$—then the self-attention can be rewritten as a basic outer product-based FWP [24, 23]. Previous works focused on different $\phi$ maps with properties such as increased capacity [23] or guaranteed approximation of the softmax in the limit [20, 21]. For our purposes, the particular choice of $\phi$ is irrelevant and we simply assume $\phi : \mathbb{R}^{d_{\text{key}}} \to \mathbb{R}^{d_{\text{key}}}$, simplifying our equations below by writing $\boldsymbol{k}, \boldsymbol{q}$ instead of $\phi(\boldsymbol{k}), \phi(\boldsymbol{q})$. Using otherwise the same notation as above, for each new input $\boldsymbol{x}^{(t)}$, the output $\boldsymbol{y}^{(t)}$ is obtained by:

$$\boldsymbol{k}^{(t)}, \boldsymbol{v}^{(t)}, \boldsymbol{q}^{(t)} = \boldsymbol{W}_k \boldsymbol{x}^{(t)}, \boldsymbol{W}_v \boldsymbol{x}^{(t)}, \boldsymbol{W}_q \boldsymbol{x}^{(t)} \qquad (5)$$

$$\boldsymbol{W}^{(t)} = \boldsymbol{W}^{(t-1)} + \boldsymbol{v}^{(t)} \otimes \boldsymbol{k}^{(t)} \qquad (6)$$

$$\boldsymbol{y}^{(t)} = \boldsymbol{W}^{(t)} \boldsymbol{q}^{(t)} \qquad (7)$$

where the slow weight matrices $\boldsymbol{W}_k \in \mathbb{R}^{d_{\text{key}} \times d_{\text{in}}}$ and $\boldsymbol{W}_v \in \mathbb{R}^{d_{\text{out}} \times d_{\text{in}}}$ are used to obtain the *key* $\boldsymbol{k}^{(t)} \in \mathbb{R}^{d_{\text{key}}}$ and the *value* $\boldsymbol{v}^{(t)} \in \mathbb{R}^{d_{\text{out}}}$. The key and value vectors are used to generate new weights via the outer product $\boldsymbol{v}^{(t)} \otimes \boldsymbol{k}^{(t)} \in \mathbb{R}^{d_{\text{out}} \times d_{\text{key}}}$. A further simplification in the equations above is the omission of attention normalisation which has been experimentally shown to be unnecessary if the $\phi$ function produces normalised key and query vectors [23].

In Eq. 6, the previous fast weight matrix $\boldsymbol{W}^{(t-1)} \in \mathbb{R}^{d_{\text{out}} \times d_{\text{key}}}$ is updated to yield $\boldsymbol{W}^{(t)}$ by adding the update term $\boldsymbol{v}^{(t)} \otimes \boldsymbol{k}^{(t)}$. This corresponds to the *sum update rule* or *purely additive programming instruction*. Here the fast NN is a simple linear transformation as in Eq. 7 which takes as input the query vector $\boldsymbol{q}^{(t)} \in \mathbb{R}^{d_{\text{key}}}$ generated by the slow weights $\boldsymbol{W}_q \in \mathbb{R}^{d_{\text{key}} \times d_{\text{in}}}$. Hence, in linear Transformers, the previous Eq. 3 simplifies to: $\boldsymbol{z}^{(t)}, \boldsymbol{q}^{(t)} = \texttt{SlowSubnet}(\boldsymbol{x}^{(t)}; \boldsymbol{\theta}_{\text{slow}})$ with $\boldsymbol{z}^{(t)} = (\boldsymbol{k}^{(t)}, \boldsymbol{v}^{(t)})$.

**Delta Net.** The Delta Net [23] is obtained by replacing the purely additive programming instruction (Eq. 6) in the linear Transformer with the one akin to the *delta rule* [33]:

$$\boldsymbol{W}^{(t)} = \boldsymbol{W}^{(t-1)} + \beta^{(t)} (\boldsymbol{v}^{(t)} - \bar{\boldsymbol{v}}^{(t)}) \otimes \boldsymbol{k}^{(t)} \qquad (8)$$

where $\beta^{(t)} \in \mathbb{R}$ is a fast parameter (learning rate) of the update rule generated by the slow net with weights $\boldsymbol{W}_\beta \in \mathbb{R}^{1 \times d_{\text{in}}}$ and the sigmoid function $\sigma$:

$$\beta^{(t)} = \sigma(\boldsymbol{W}_\beta \boldsymbol{x}^{(t)}) \qquad (9)$$

and $\bar{\boldsymbol{v}}^{(t)} \in \mathbb{R}^{d_{\text{out}}}$ is generated as a function of the previous fast weights $\boldsymbol{W}^{(t-1)}$ and the key $\boldsymbol{k}^{(t)}$

$$\bar{\boldsymbol{v}}^{(t)} = \boldsymbol{W}^{(t-1)} \boldsymbol{k}^{(t)}. \qquad (10)$$

This update rule was introduced to address a memory capacity problem affecting linear Transformers with the purely additive update rule [23]. The corresponding Eq. 3 is: $\boldsymbol{z}^{(t)}, \boldsymbol{q}^{(t)} = $ SlowSubnet$(\boldsymbol{x}^{(t)}, \boldsymbol{W}^{(t-1)}; \boldsymbol{\theta}_{\text{slow}})$ with $\boldsymbol{z}^{(t)} = (\boldsymbol{k}^{(t)}, \boldsymbol{v}^{(t)}, \beta^{(t)}, \bar{\boldsymbol{v}}^{(t)})$. Thus, unlike linear Transformers, the SlowNet in the Delta Net takes the previous fast weights $\boldsymbol{W}^{(t-1)}$ into account to generate the new fast weight updates.

We typically use the multi-head version [1] of the computations above. After the projection (Eq. 5), the vectors $\boldsymbol{k}^{(t)}, \boldsymbol{v}^{(t)}, \boldsymbol{q}^{(t)}$ are split into equally sized $H$ sub-vectors, and the rest of the operations are conducted by $H$ computational heads independently. The resulting output vectors from each head are concatenated to form the final output.

**Other approaches.**    While our focus here is on outer product-based weight generation, which is an efficient method to handle high dimensional NN weights, there are also other approaches. For example, instead of generating a new weight matrix, Hypernetworks [34] scale the rows of a slow weight matrix with a generated vector of appropriate size. Weight compression to control fast weights in a low dimensional compressed space has been also studied [35]. In the broad sense of context-dependent weights [36, 37, 38], many concepts relate to FWPs: e.g. dynamic convolution [39, 40, 41], LambdaNetworks [42], or dynamic plasticity [43, 44].

## 3   Fast Weight Programmers With Slow or Fast RNNs

The original formulation of FWPs reviewed in Sec. 2.1 is more general than existing models presented in Sec. 2.2. In particular, both fast and slow networks in existing linear Transformers consist of a single feedforward layer (Eqs. 5 and 7). Here we present FWPs with recurrent fast nets in Sec. 3.1 and FWPs with recurrent slow nets in Sec. 3.2.

### 3.1   Fast Network Extensions

In principle, any NN architecture can be made *fast*. Its fast weight version is obtained by replacing the networks' weights with fast weights parameterised by an additional slow network. For example, consider a regular RNN layer with two weight matrices $\boldsymbol{W}$ and $\boldsymbol{R}$:

$$\boldsymbol{h}^{(t)} \quad = \quad \sigma(\boldsymbol{W}\boldsymbol{x}^{(t)} + \boldsymbol{R}\boldsymbol{h}^{(t-1)}) \tag{11}$$

A fast weight version can be obtained by replacing $\boldsymbol{W}$ and $\boldsymbol{R}$ with $\boldsymbol{W}^{(t)}$ and $\boldsymbol{R}^{(t)}$ which are controlled as in Eq. 8 with all necessary variables generated by a separate slow net at each time step $t$.

While this view illustrates the generality of FWPs, the angle under which we approach these models is slightly different: we introduce recurrence as a way of augmenting existing linear Transformers.

**Delta RNN.**    We obtain a fast weight RNN called **Delta RNN** by adding an additional recurrent term to the feedforward fast net of the linear Transformer (Eq. 7):

$$\boldsymbol{y}^{(t)} \quad = \quad \boldsymbol{W}^{(t)}\boldsymbol{q}^{(t)} + \boldsymbol{R}^{(t)}f(\boldsymbol{y}^{(t-1)}) \tag{12}$$

where $\boldsymbol{R}^{(t)} \in \mathbb{R}^{d_{\text{out}} \times d_{\text{out}}}$ is an additional fast weight matrix which introduces recurrent connections. It is also generated by the slow net using the delta update rule, similar to $\boldsymbol{W}^{(t)}$ in Eq. 8 but with additional slow weights. We apply an element-wise activation function $f$ to the previous output of the fast network $\boldsymbol{y}^{(t-1)}$ to obtain the recurrent query. The choice of activation function is crucial here because, to achieve stable model behaviour, the elements in key and query vectors should be positive and sum up to one when the delta update rule is used [23]. We use the softmax function ($f = $ softmax in Eq. 12) to satisfy these conditions. An ablation study on the choice of using Eq. 12 instead of the one similar to Eq. 11 can be found in Appendix A.2.

Analogous to the Delta RNN, we also construct a **Delta LSTM** with six fast weight matrices. The exact equations can be found in Appendix A.2.

**Alternative Feedforward Fast Nets.**    While the focus of this work is on RNNs, there are also interesting fast feedforward models to be used in Eq. 7 which might result in stronger feedforward

baselines. For example, we can replace the single layer fast net of Eq. 7 by a $K$-layer deep network:

$$\boldsymbol{h}_k^{(t)} = \boldsymbol{W}_k^{(t)} f(\boldsymbol{h}_{k-1}^{(t)}) \quad \text{for } k \in [1..K] \text{ with } \boldsymbol{h}_0^{(t)} = \boldsymbol{q}^{(t)} \tag{13}$$

$$\boldsymbol{y}^{(t)} = \boldsymbol{h}_K^{(t)} \tag{14}$$

where the slow network produces all $K$ fast weights $\{\boldsymbol{W}_k^{(t)}\}_{k=1}^K$ and query $\boldsymbol{q}^{(t)}$ from a single input $\boldsymbol{x}^{(t)}$. In light of the capacity limitation in linear Transformers [23], this might introduce additional capacity without the need of larger representations, analogous to the trade-off in a multilayer perceptron (MLP) between narrow & deep versus shallow & wide. We refer to this class of models as **Delta MLPs**. Again, for stable model behaviour with the delta rule, we apply the softmax activation $f$ to the vectors to be used as a query.

Another interesting approach is to use a Delta Net itself as a fast net, i.e., make the slow weights in the Delta Net fast (thus obtaining a **Delta Delta Net**). Such a model could in principle learn to adapt the way of generating fast weights depending on the context. While we plan to investigate the potential of such hierarchical FWPs in future work, we also include preliminary results of such a model in our language modelling experiments (Sec. 4.1). A discussion on the dimensionality of such a model can also be found in Appendix A.3.

We experimentally demonstrate that (slow) NNs can learn to control the weights of these rather complex fast networks (Sec. 4).

### 3.2 Slow Network Extensions

In linear Transformers, the slow network is purely feedforward (Eq. 5). It can be made recurrent at two different levels: within the slow network (i.e. the slow network computes weight updates based on its own previous outputs e.g., key, value, query vectors) or via the fast network by taking the fast net's previous output as an input. In our preliminary experiments, we found the former to be sub-optimal (at least in language modelling experiments). So we focus on the latter approach: we make the slow net in the Delta Net dependent on the previous output of the fast network. We refer to this model as the **Recurrent Delta Net** (RDN).

**Recurrent Delta Net.** We obtain the RDN by modifying the generation of key, value, and query vectors (Eq. 5) as well as the learning rate (Eq. 9) in the Delta Net. We add additional slow weights $(\boldsymbol{R}_k, \boldsymbol{R}_q \in \mathbb{R}^{d_{\text{key}} \times d_{\text{out}}}, \boldsymbol{R}_v \in \mathbb{R}^{d_{\text{out}} \times d_{\text{out}}}$, and $\boldsymbol{R}_\beta \in \mathbb{R}^{1 \times d_{\text{out}}})$ for recurrent connections which connect the previous output of the fast net $\boldsymbol{y}^{(t-1)}$ (Eq. 7) to the new $\boldsymbol{k}^{(t)}, \boldsymbol{v}^{(t)}, \boldsymbol{q}^{(t)}$, and $\beta^{(t)}$ as follows:

$$\boldsymbol{k}^{(t)} = \boldsymbol{W}_k \boldsymbol{x}^{(t)} + \boldsymbol{R}_k \tanh(\boldsymbol{y}^{(t-1)}) \tag{15}$$

$$\boldsymbol{v}^{(t)} = \boldsymbol{W}_v \boldsymbol{x}^{(t)} + \boldsymbol{R}_v \tanh(\boldsymbol{y}^{(t-1)}) \tag{16}$$

$$\boldsymbol{q}^{(t)} = \boldsymbol{W}_q \boldsymbol{x}^{(t)} + \boldsymbol{R}_q \tanh(\boldsymbol{y}^{(t-1)}) \tag{17}$$

$$\beta^{(t)} = \sigma(\boldsymbol{W}_\beta \boldsymbol{x}^{(t)} + \boldsymbol{R}_\beta \tanh(\boldsymbol{y}^{(t-1)})) \tag{18}$$

While the rest of the model remains as in the Delta Net, with these simple extra recurrent connections the model becomes a proper RNN. The corresponding dependencies in Eq. 3 are: $\boldsymbol{z}^{(t)}, \boldsymbol{q}^{(t)} = \texttt{SlowSubnet}(\boldsymbol{x}^{(t)}, \boldsymbol{y}^{(t-1)}, \boldsymbol{W}^{(t-1)}; \boldsymbol{\theta}_{\text{slow}})$ with $\boldsymbol{z}^{(t)} = (\boldsymbol{k}^{(t)}, \boldsymbol{v}^{(t)}, \beta^{(t)}, \bar{\boldsymbol{v}}^{(t)})$.

### 3.3 Related Models

All the RFWP models presented in Sec. 3.1 and 3.2 can be seen as a type of memory augmented recurrent neural networks [45, 46] in the sense that they maintain two-dimensional fast weight states as a short-term memory, in addition to the standard one-dimensional RNN states.

There are also several previously proposed recurrent fast weight models. For example, Schmidhuber's recurrent FWP from 1993 [26] has been revisited by Ba et al. [47]. There, key and value vectors are not generated within the same time step, unlike in our models or in linear Transformers. The Fast Weight Memory (FWM) [48] is also a recurrent FWP: the slow net is an LSTM and the fast net is a higher-order RNN. However, the FWM is a single pair of slow and fast nets, and a multi-layer version, as in the linear Transformer family, was not explored. Similarly, the Metalearned Neural Memory [49] uses an LSTM as its slow net and a 3-layer MLP as its fast net but again limited to one pair.

Table 1: WikiText-103 language model perplexity results with the *small* setting [21, 23]. For each model, its name, corresponding slow and fast networks, and weight update rule (Update) are specified. All models are trained and evaluated on the span of 256 tokens except for the models in the last two rows (+ full context) which are trained and evaluated without context truncation. Parameter count is in millions. See Appendix A for further experimental details and results.

| Name | Slow net | Update | Fast net | Valid | Test | #Prms |
|------|----------|--------|----------|-------|------|-------|
| Transformer | - | - | - | 33.0 | 34.1 | 44.0 |
| Linear Transformer | Feedforward | sum | Linear | 37.1 | 38.3 | 44.0 |
| Delta Net | | delta | | 34.1 | 35.2 | 44.0 |
| Delta MLP | Feedforward | delta | Deep MLP | 35.8 | 36.8 | 44.3 |
| Delta Delta Net | | | Delta Net | 34.0 | 35.2 | 44.6 |
| Delta RNN | | | RNN | 33.8 | 35.0 | 44.6 |
| Delta LSTM | | | LSTM | **32.6** | **33.8** | 47.3 |
| RDN | Recurrent | | Linear | 34.1 | 35.2 | 44.1 |
| Delta RNN | + full context | | | **31.8** | **32.8** | 44.6 |
| RDN | | | | 32.5 | 33.6 | 44.1 |

Others have investigated variants of RNNs with fast weights for toy synthetic retrieval tasks [50, 51]. In particular, Keller et al. [51] augment the LSTM with a fast weight matrix in the cell update. In contrast, we make all weights in the LSTM fast and, importantly, our model specifications build upon the successful deep Transformer architecture using residual connections [52, 53], layer-norm [54], multiple attention heads and feed-forward blocks [1]. Essentially, we replace the self-attention layers in the regular Transformers by the fast weight programmer operations described above.

## 4   Experiments

We conduct experiments in four different settings. We start by evaluating all models on a language modelling task (Sec. 4.1) to obtain a performance overview and to discuss computational costs. Language modelling is an excellent task to evaluate sequence models. However, to highlight their different capabilities, we evaluate our models also on algorithmic tasks. In fact, it is well-known that the actual capabilities of RNNs differ from one architecture to another [55]. We are interested in discussing such differences. With that goal in mind, we conduct experiments on two synthetic algorithmic tasks, code execution (Sec. 4.2) and sequential ListOps (Sec. 4.3), which are designed to compare elementary sequence processing abilities of models. Finally, we apply our models to reinforcement learning in 2D game environments (Sec. 4.4) as a replacement for LSTMs.

### 4.1   Language Modelling

We first evaluate all discussed models on the generic language modelling task. This allows for obtaining a performance overview and reviewing the computational efficiency of different models. We use the Wikitext-103 dataset [28] and follow the *small model setting* similar to what's used in recent works by Peng et al. [21] and Schlag et al. [23]. This allows for training and evaluating different models with a reasonable amount of compute on this resource-demanding language modelling task.

**Perplexity results.**   The results are shown in Table 1 which also serves as a tabular summary recapitulating different models described in Sec. 2 and 3, with various architectures for slow and fast nets, and two choices of update rule. The top block of Table 1 shows the performance of the baseline Transformer, Katharopoulos et al. [19]'s Linear Transformer, and Schlag et al. [23]'s Delta Net. The performance of models presented in Sec. 3 can be found in the middle block. First of all, the Delta MLP performs worse than the baseline Delta Net despite a slight increase in parameter count (44.3 vs. 44.0 M). This supports the intuition that it is better to make the slow network aware of the outputs of intermediate layers to generate fast weights in a deep network, instead of generating fast weights for all layers at a time. In all other models, the performance never degrades with the proposed architectural augmentation. The Delta Delta Net yields limited improvements; we plan to study this

model in depth in future work. With the same amount of parameters (44.6 M), the Delta RNN yields greater improvements. Among the models presented here, the Delta LSTM variant exhibits the best performance. This shows that the slow network successfully controls the rather complex fast LSTM network, although it also requires more parameters (47.3 M) than other models. Finally, the benefits of recurrent connections added to the baseline Delta Net do not directly translate into practical improvements in language modelling as demonstrated by the performance of RDN compared to the one of the baseline Delta Net. Importantly, given a constant memory size w.r.t. sequence length, it is straight-forward to train and evaluate our RNNs without context truncation (while still limiting the backpropagation span). Corresponding performances of Delta RNN and RDN are shown in the bottom part of Table 1: they outperform the regular Transformer with a limited context (256 tokens).

While language modelling is useful as a sanity check (here for example, except for the Delta MLP, all models achieve reasonable performance), the task is too generic to identify certain important aspects of the models, such as real benefits of recurrence. Before we move on to trickier RL applications, Sec. 4.2 and 4.3 will focus on studying such aspects using synthetic algorithmic tasks.

**Computational efficiency.** The modifications we proposed in Sec. 3 introduce additional computational costs to linear Transformers/FWPs. First of all, none of them affect the core complexity of linear Transformers: they all have a constant space and linear time complexity w.r.t. sequence length. However, the per-time-step computational costs differ a lot from one model to another, as quantified here in terms of training speed using our implementation. All models are implemented using a custom CUDA kernel except the baseline Transformer for which we use regular PyTorch code [56]. Training speeds of LT and Delta Net in Table 1 are 66 K and 63 K words per second respectively (vs. 33 K for the baseline Transformer). The most expensive model is the Delta LSTM. This fast weight LSTM with tied input-forget gates has 6 weight matrices, and each of these are manipulated by separate delta rules. The corresponding speed is 14 K words per second, too slow for scaling to more experiments. In contrast, the speeds of Delta RNN and RDN remain reasonable: 41 K and 35 K words per second respectively. Therefore, the remaining experiments will focus on these two recurrent architectures which are promising and practical in terms of both performance and computational costs.

## 4.2 Code Execution Task: Learning to Maintain and Update Variable States

In code execution tasks [29], models are trained to sequentially read the input code provided as word-level text, and to predict the results of the corresponding code execution. We adopt the task setting from Fan et al. [57] with one conditional and three basic statements. We refer the readers to Appendix B.1 for a precise description of the task. This code execution task requires models to maintain the values of multiple variables, which has been shown to be difficult for relatively shallow Transformers with only feedforward connections [57].

The left block of Table 2 shows the results. Following again Fan et al. [57], we control the task difficulty by modifying the number of variables (3 or 5). The model architectures are fixed: the LSTM has only one layer with 256 nodes and all Transformer variants have the same architecture with 4 layers with a hidden size of 256 using 16 heads and an inner feedforward layer size of 1024.

We first note that the LSTM is the best performer for both difficulty levels, with the smallest performance drops through increasing the number of variables. In contrast to prior claims [57], the LSTM is clearly capable of storing the values of multiple variables in its hidden and cell state vectors. With three variables, the regular Transformer already largely underperforms other models with a mutable memory: Delta Net, Delta RNN, and RDN. Linear Transformers completely fail at this task, likely due to the memory capacity problem pointed out by Schlag et al. [23] (see Appendix B.2 for further discussion). By increasing the number of variables to five, the baseline Transformers, Delta Net, and RDN become unstable as shown by high standard deviations w.r.t. the seed. The benefits of recurrent connections introduced in our RDN compared to the baseline Delta Net become more apparent (76.3 vs. 61.4%). In contrast, the Delta RNN remains stable and gives the best performance (85.1%) among Transformer variants, which shows the benefits of recurrence and in particular the regular RNN architecture in the fast net. To match the performance of LSTM on this task, however, these models need more layers (see Appendix B.2 for more results).

Table 2: **Test accuracies** (%) with standard deviations on **code execution** (Code Exec) and **sequential ListOps** (Seq ListOps). The difficulty of the task is controlled by the maximum number of possible variables (# variables) for code execution, and the list depth (10 or 15) for ListOps. For code execution with 5 variables, we report means over six seeds. In all other cases, the results are computed with three seeds. For more results, see Appendix B.2 (Code Exec) and B.4 (Seq ListOps).

|  | Code Exec (# variables) | | Seq ListOps (depth) | |
|---|---|---|---|---|
|  | 3 | 5 | 10 | 15 |
| LSTM | **99.0** ± 0.1 | **93.2** ± 6.1 | **88.5** ± 2.9 | 24.4 ± 1.1 |
| Transformer | 71.8 ± 2.6 | 35.4 ± 28.2 | 79.1 ± 0.9 | 75.3 ± 0.4 |
| Linear Transformer | 0.0 ± 0.0 | 0.0 ± 0.0 | 64.0 ± 0.3 | 64.4 ± 0.4 |
| Delta Net | 90.7 ± 2.7 | 61.4 ± 20.0 | **85.7** ± 1.8 | 77.6 ± 1.4 |
| Delta RNN | 90.8 ± 1.7 | **85.1** ± 1.9 | 83.6 ± 1.2 | 78.0 ± 1.0 |
| RDN | **92.6** ± 2.2 | 76.3 ± 17.6 | 83.2 ± 0.9 | **79.2** ± 1.4 |

## 4.3 Sequential ListOps: Learning Hierarchical Structure and Computation

The ListOps task [30] is a typical test for hierarchical structure learning, which requires list operation executions. We use a simple variant of ListOps whose detailed descriptions can be found in Appendix B.4. For example, the list [MAX 6 1 [FIRST 2 3 ] 0 [MIN 4 7 1] ] is of depth two and the expected output is 6. While early research comparing self-attention to RNNs [58] has shown some advantages of recurrence in hierarchical structure learning, more recent work [59] reports Transformers outperforming LSTMs on ListOps. According to Tay et al. [22], linear Transformer variants (LT and Performers) underperform other Transformer variants by a large margin on ListOps.

The right block of Table 2 shows results for two different depths: 10 and 15. The model architectures are identical to those used in the code execution task (Sec. 4.2). At depth 10, we find LSTM to perform best, while mutable memory Transformer variants (Delta Net, Delta RNN, and RDN) outperform the regular and linear Transformers. At depth 15, the LSTM's performance drops drastically (to 24.4%), while the differences between Transformer variants remain almost the same. We note that sequences are longer for the depth 15 problem (mean length of 185 tokens) than for the depth 10 version (mean length of 98 tokens). This turns out to be difficult for the small 256-dimensional LSTM; see Appendix B.4 for the corresponding ablation study. The performance differences between the baseline Delta Net and the proposed Delta RNN and RDN are rather small for this task. Importantly, our models outperform both regular and linear Transformers on this task requiring hierarchical structure learning.

## 4.4 Reinforcement Learning in 2D Game Environments

We finally evaluate the performance of our models as a direct replacement for the LSTM in reinforcement learning settings. In fact, only a limited number of prior works have investigated Transformers for RL. Parisotto et al. [12] and Rae et al. [11] evaluate them on the DMLab-30 [60, 61]. Parisotto et al. [12] also evaluate them on Atari but in a multi-task setting [62]. Others [57, 13] use toy maze environments. In contrast to Parisotto et al. [12]'s work, which presents multi-task Atari as a side experiment, we study the Transformer family of models on the standard Atari 2600 setting [31, 63, 64] by training game-specific agents.

**Settings.** We train an expert agent on each game separately with the Importance Weighted Actor-Learner Training Architecture (IMPALA) using the V-trace actor-critic setup [65] and entropy regularization [66] implemented in `Torchbeast` [67]. Our model follows the *large* architecture of Espeholt et al. [65] which consists of a 15-layer residual convolutional NN with one 256-node LSTM layer which we replace by either the RDN (Sec. 3.2) or the Delta RNN (Sec. 3.1). In line with the small LSTM used for Atari (only 1 layer with 256 hidden nodes) we also configure a small RDN: 2 layers with a hidden size of 128 using 4 heads, and a feedforward dimension of 512. We find this small model to perform already surprisingly well. For the rest, we use the same hyperparameters as Espeholt et al. [65] which can be found in Appendix C.

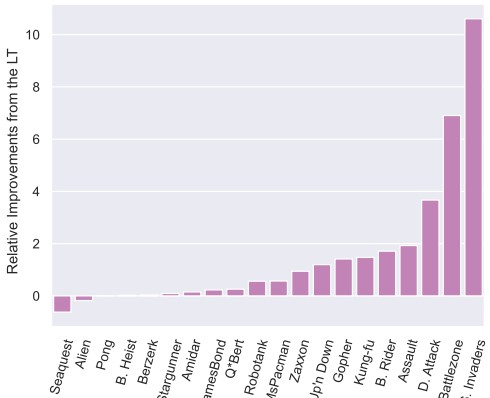
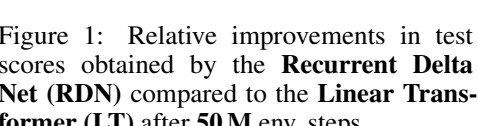

Figure 1: Relative improvements in test scores obtained by the **Recurrent Delta Net (RDN)** compared to the **Linear Transformer (LT)** after **50 M** env. steps.

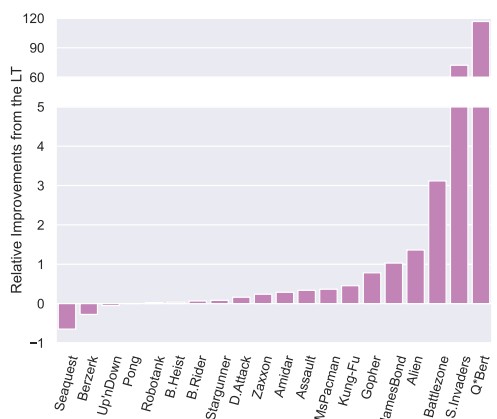

Figure 2: Relative improvements in test scores obtained by the **Recurrent Delta Net (RDN)** compared to the **Linear Transformer (LT)** after **200 M** env. steps.

**Main experiments.** We evaluate our models in 20 environments. According to Mott et al. [68], in about half of them, the LSTM outperforms the feedforward baselines—which we confirm in our setting with 50 M steps (see Appendix C). We report results at 50 M and 200 M environmental steps of training. Like Nair et al. [69], we run the trained agent for 30 test episodes. Here we repeat this evaluation five times to report the average score with a standard deviation. The following analysis focuses on the RDN (Sec. 3.2) compared to the regular linear Transformer and the LSTM. A similar study of the Delta RNN, as well as comparisons to more baselines, and the exact scores achieved by each model on each game can be found in Appendix C.

In all our experiments above, we have shown that the Linear Transformer, i.e., a Fast Weight Programmer with a purely additive update rule, consistently underperforms other models based on the delta rule. Here we confirm this trend once more. Figures 1 and 2 show the relative improvements of scores obtained by Recurrent Delta Net over those achieved by the linear Transformer on each game, respectively after 50 and 200 M interaction steps. The RDN matches or outperforms the Linear Transformer on all games except for two out of 20 games at both stages of training.

Figure 3 shows relative improvements of RDN over LSTM after 50 M interactions. In 12 games, the RDN yields improvements over LSTM, whereas in 3 games, the LSTM performs better. In the

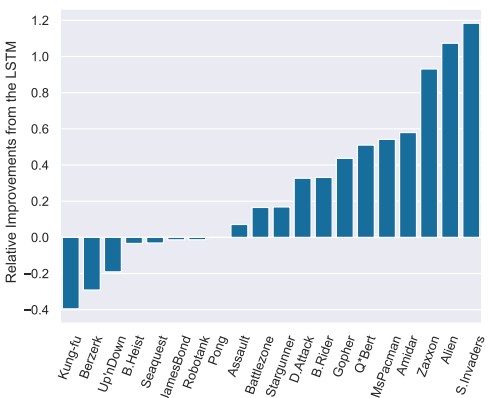

Figure 3: Relative improvements in test scores obtained by 2-layer **RDN** compared to **LSTM** after **50 M** env. steps.

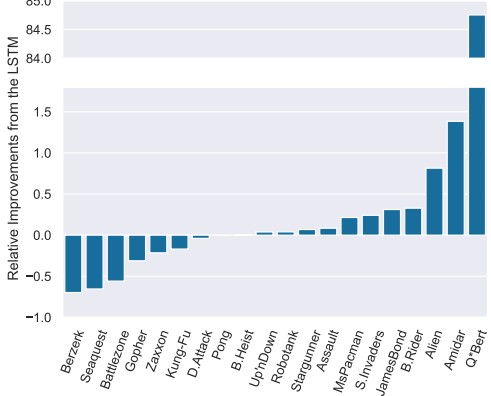

Figure 4: Relative improvements in test scores obtained by 2-layer **RDN** compared to **LSTM** after **200 M** env. steps.

remaining 5 games, both reach similar scores. Interestingly, this trend does not directly extrapolate to the 200 M case, which is presented in Figure 4. With longer training, the LSTM surpasses the performance of the RDN in *Battlezone*, *Gopher*, *Seaquest* and *Zaxxon*, while the RDN catches up in *Up'N Down* and *Kung-Fu Master*. Overall, there are 6 games in which LSTM clearly outperforms RDN at 200 M steps, whereas in 9 games the result is the opposite.

On a side note, some of the scores achieved by the RDN at 200 M step are excellent: a score of over 170 K and 980 K in *Space Invader* and *Q\*Bert* respectively beats the state-of-the-art set by MuZero [70] and Agent57 [62]. However, a direct comparison is not fair as we train game-specific agents.

**Experiments with larger models.**   Given the results above, a natural question to ask is whether a larger model size improves the RDN in games where the LSTM dominates. We focus on four such games: *Battlezone*, *Berzerk*, *Gopher*, and *Seaquest* (See Fig. 4). We double the model size to 3.4 M parameters by increasing the number of layers to 4 and the hidden size to 256, with 8 heads. As shown in Table 3, larger RDN models reduce the gap to the LSTM (except in *Berzerk*). This indicates that further scaling RDN might be as promising as scaling regular Transformers in other domains.

Table 3: Performance of a larger RDN in **games where the LSTM dominates** (200 M steps).

|  | Battlezone | Berzerk | Gopher | Seaquest |
|---|---|---|---|---|
| LSTM | $24,873 \pm 1,240$ | $\mathbf{1,150} \pm 92$ | $\mathbf{124,914} \pm 22,422$ | $12,643 \pm 1,627$ |
| RDN | $10,980 \pm 1,104$ | $348 \pm 17$ | $86,008 \pm 11,815$ | $4,373 \pm 504$ |
| RDN larger | $\mathbf{28,273} \pm 5,333$ | $346 \pm 9$ | $118,273 \pm 14,872$ | $\mathbf{14,601} \pm 712$ |

## 5   Conclusion

Inspired by the formal equivalence of linear Transformers and certain traditional Fast Weight Programmers (FWPs) from the early '90s, we propose various new linear Transformer variants with recurrent connections. Our novel Recurrent FWPs (RFWPs) outperform previous linear and regular Transformers on a code execution task and significantly improve over Transformers in a sequential ListOps task. On Wikitext-103 in the "small" model setting, RFWPs compete well with the previous best linear Transformer variants for truncated contexts, and with full contexts, beat regular Transformers. Our RFWPs can also be used as drop-in replacements for problems where RNNs are still dominant. In particular, we evaluate them in reinforcement learning settings on 20 Atari 2600 environments. They clearly outperform the regular Linear Transformer on almost all environments. They also outperform the LSTM across many environments with a small model size and demonstrate promising scaling properties for larger models. Given the increasing interest in deploying Transformers in RL [71, 72], in particular in the framework of Upside-Down RL [73, 74], our RFWP models are particularly relevant: as RNNs, they conveniently handle long contexts with a constant memory size, while being powerful Transformer variants at the same time. Our work highlights the usefulness of the FWP framework from the '90s and its connection to modern architectures, opening promising avenues for further research into new classes of recurrent Transformers.

## Acknowledgments and Disclosure of Funding

We thank Aleksandar Stanić and Sjoerd van Steenkiste for valuable comments on the first version of this paper. This research was partially funded by ERC Advanced grant no: 742870, project AlgoRNN, and by Swiss National Science Foundation grant no: 200021_192356, project NEUSYM. This work was partially supported by computational resources at the CSCS Swiss National Supercomputing Centre, project d115. We thank NVIDIA Corporation for donating several DGX machines, and IBM for donating a Minsky machine.

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
