# A Experimental Details and Ablation Studies for Language Modelling

## A.1 Experimental Settings

All language models in Table 1 have the same Transformer configuration: a 16-layer model with a hidden size of 128 with 8 heads, and a feed-forward dimension of 2048. We use a dropout [75, 76, 77] rate of 0.1. The batch size is 96 and we train for about 120 epochs with Adam optimiser [78] with an initial learning rate of 0.00025 and 2000 learning rate warm-up steps. All models are trained with a back-propagation span of 256 tokens. During training, these segments are treated independently, except for the + *full context* cases in Table 1 where the states (both recurrent states and fast weight states) from a segment are used as initialisation for the subsequent segment. The models in + *full context* cases are also evaluated in the same way by carrying over the context throughout the evaluation text with a batch size of one. For all other cases, the evaluation is done by going through the text with a sliding window of size 256 with a batch size of one. Transformer states are computed for all positions in each window, but only the last position is used to compute perplexity (except in the first segment where all positions are used for evaluation) [2]. We trained all models using two GPUs (32 GB V100), and the longest training takes up to 10 days (see Sec. 4.1 in the main text for speed comparison between models).

For readers interested in any further details, we refer to our code which is publicly available.

## A.2 Ablation Studies

In this section, we specify the exact Delta LSTM and Delta MLP models used in Table 1, and provide a few ablation studies for Delta RNN, Delta LSTM[3], and Delta MLP models.

Table 4: Ablation studies for Delta LSTM, Delta RNN, and Delta MLP models. Language model perplexities are shown and the setting is the same as in Table 1.

|            | Version | Valid | Test  | #Prms |
|------------|---------|-------|-------|-------|
| Delta RNN  | A       | 35.6  | 36.7  | 44.6  |
|            | B       | **33.8** | **35.0** |       |
| Delta LSTM | A       | 38.5  | 39.9  | 47.3  |
|            | B       | 34.2  | 35.2  |       |
|            | C       | 33.5  | 34.7  |       |
|            | D       | **32.6** | **33.8** |       |
| Delta MLP  | A       | 36.8  | 37.9  | 44.3  |
|            | B       | **35.8** | **36.8** | 44.3  |

**Delta RNN.** In Sec. 3.1, we argue for a version of fast RNN given by Eq. 12 as a natural augmentation of the linear Transformer with recurrent connections. Here we empirically support this choice by comparing to another variant of Delta RNN given by:

$$\boldsymbol{y}^{(t)} \quad = \quad f(\boldsymbol{W}^{(t)}\boldsymbol{q}^{(t)} + \boldsymbol{R}^{(t)}\boldsymbol{y}^{(t-1)}) \tag{19}$$

where $f$ is again the softmax activation which makes $\boldsymbol{y}^{(t)}$ a valid query vector (positive components which sum up to one) for fast weights maintained by the delta update rule. We refer to this version as **Version A** in this ablation study and the one given by Eq. 12 as **Version B**. As Table 4 shows, Version B performs better, and this is the one we report in Table 1 in the main text.

**Delta LSTM.** We evaluate four versions of Delta LSTM for ablation. In all cases, we tie the input and forget gates to reduce the total number of fast weights to be controlled by the slow net. All

---

[3]The numbers reported in Table 4 for the Delta LSTM models are better than those we presented in an earlier version. In fact, we found that in our previous code, the slow weights for key and value generation were shared by mistake between the forward and recurrent fast weight matrices (while the reported parameter count was that of the correct model with separate slow weight matrices). Fixing this resulted in the corresponding improvements.

models contain six fast weights and each of them is updated according to the delta update rule (Eq. 8). The different versions differ in the location of the activation function and residual connections in the LSTM architecture [14, 79], inspired by the Transformer.

**Version A** (analogous to Version A of Delta RNN above) is the one which is the closest to the original LSTM with tied input and forget gate. The only architectural difference is the usual $\texttt{tanh}$ on the cell output $c^{(t)}$ which is replaced by a softmax $f$ placed after the final output of the layer $f(c^{(t)} \odot o^{(t)})$, such that it can directly be used as a delta rule compatible query for the next time step (we also use a sigmoid instead of $\texttt{tanh}$ for the main transformation $u^{(t)}$, but this is not crucial for any models here).

$$u^{(t)} = \sigma(W^{(t)}q^{(t)} + R^{(t)}y^{(t-1)}) \tag{20}$$

$$f^{(t)} = \sigma(W_f^{(t)}q^{(t)} + R_f^{(t)}y^{(t-1)}) \tag{21}$$

$$o^{(t)} = \sigma(W_o^{(t)}q^{(t)} + R_o^{(t)}y^{(t-1)}) \tag{22}$$

$$c^{(t)} = f^{(t)} \odot c^{(t-1)} + (1 - f^{(t)}) \odot u^{(t)} \tag{23}$$

$$y^{(t)} = f(c^{(t)} \odot o^{(t)}) \tag{24}$$

**Version B** is obtained by delaying the application of the softmax activation $f$ in Version A.

$$u^{(t)} = \sigma(W^{(t)}q^{(t)} + R^{(t)}f(y^{(t-1)})) \tag{25}$$

$$f^{(t)} = \sigma(W_f^{(t)}q^{(t)} + R_f^{(t)}f(y^{(t-1)})) \tag{26}$$

$$o^{(t)} = \sigma(W_o^{(t)}q^{(t)} + R_o^{(t)}f(y^{(t-1)})) \tag{27}$$

$$c^{(t)} = f^{(t)} \odot c^{(t-1)} + (1 - f^{(t)}) \odot u^{(t)} \tag{28}$$

$$y^{(t)} = c^{(t)} \odot o^{(t)} \tag{29}$$

**Version C** is obtained from Version B by adding a residual connection from the feed-forward part $z_u^{(t)}$ of the main transformation $u^{(t)}$ to the output.

$$z_u^{(t)} = W^{(t)}q^{(t)} \tag{30}$$

$$u^{(t)} = \sigma(z_u^{(t)} + R^{(t)}f(y^{(t-1)})) \tag{31}$$

$$f^{(t)} = \sigma(W_f^{(t)}q^{(t)} + R_f^{(t)}f(y^{(t-1)})) \tag{32}$$

$$o^{(t)} = \sigma(W_o^{(t)}q^{(t)} + R_o^{(t)}f(y^{(t-1)})) \tag{33}$$

$$c^{(t)} = f^{(t)} \odot c^{(t-1)} + (1 - f^{(t)}) \odot u^{(t)} \tag{34}$$

$$y^{(t)} = c^{(t)} \odot o^{(t)} + z_u^{(t)} \tag{35}$$

Finally, **Version D** is obtained from Version B by removing the sigmoid on the main transformation $u^{(t)}$ which results in a highway net-like skip connection [53] from $u^{(t)}$ to the output. This version is then analogous to Version B of the Delta RNN as a natural augmentation of the linear Transformer: a recurrent term is added to the main transformation $u^{(t)}$ and gating components are added to make it an LSTM architecture:

$$u^{(t)} = W^{(t)}q^{(t)} + R^{(t)}f(y^{(t-1)}) \tag{36}$$

$$f^{(t)} = \sigma(W_f^{(t)}q^{(t)} + R_f^{(t)}f(y^{(t-1)})) \tag{37}$$

$$o^{(t)} = \sigma(W_o^{(t)}q^{(t)} + R_o^{(t)}f(y^{(t-1)})) \tag{38}$$

$$c^{(t)} = f^{(t)} \odot c^{(t-1)} + (1 - f^{(t)}) \odot u^{(t)} \tag{39}$$

$$y^{(t)} = c^{(t)} \odot o^{(t)} \tag{40}$$

Corresponding performances can be found in Table 4. The best model, Version D, is the one we report in Table 1 in the main text.

**Delta MLP.** We also conduct a few ablation studies for the Delta MLP (Sec. 3.1). As MLP architecture we used the feedforward block of the regular Transformer which consists of two feedforward

layers: one with the size of the inner feedforward layer (2048 here) and another one with the size of hidden dimension (128 here). We test two variants which result in a similar number of parameters: **Version A** with 8 overall Transformer layers where each self-attention layer contains 4 fast MLP layers (i.e. a total of 48 feedforward layers with 32 fast ones), and **Version B** with 11 overall Transformer layers where each self-attention layer contains 2 fast MLP layers (i.e. a total of 44 feedforward layers with 22 fast ones). As shown in Table 4, Version B which has fewer fast layers controlled by the same slow net performs better, and, as already mentioned in Sec. 4, they do not outperform the baseline Delta Net which has only one fast feedforward layer (Table 1).

### A.3 Dimensionality of Delta-Delta Net vs. Delta Net

Here we describe how the dimensionality of Delta-Delta Net scales with the size of the Delta Net. We assume a Delta Net with a dimension $d$ for all query, key, value and input vectors. Then its slow weight matrix (the projection matrix) is of size $d \times (3d + 1)$ as it projects a $d$-dimensional input to query, key, value vectors ($3d$) and a scalar beta ($+1$) which are needed to maintain a $d \times d$ fast weight matrix using the delta rule. Now we can express the dimensionality of a Delta-Delta Net in terms of $d$, whose fast network is a Delta Net with the dimensionality above. The size of its fast weight matrix is thus $d \times (3d + 1)$. In order to maintain a fast weight matrix of this dimension using the delta rule, we need key and query vectors of size $d$, a value vector of size $3d + 1$, and a scalar beta ($+1$). The slow weight matrix has to produce all these variables with a total dimension of $(5d + 2)$ from the input of size $d$. Therefore, the size of the slow weight matrix in the Delta-Delta Net is $d \times (5d+2)$. Such a Delta-Delta Net would have to store two fast weight matrices: one of size $d \times (3d+1)$ and another one of size $d \times d$.

## B  Experimental Details and Additional Results for Algorithmic Tasks

### B.1  Task Details for Code Execution

In code execution tasks [29], models are trained to sequentially read the input code provided as word-level text and to predict the results of the corresponding code execution. We adopt the task setting from Fan et al. [57]. Each example is a sequence consisting of multiple statements — 100 in our experiments. A statement can be one of the following three basic statements: `assign` which assigns a value to a variable (e.g. `x = 2 ;`), `increment` which increments or decrements an already assigned variable (e.g. `x ++ ;`), or `print` which outputs the value of the variable (e.g. `print x ;`). In addition to basic statements, there are also conditional comparisons on already defined variables followed by a basic statement (e.g. `if x < 3 : x ++ ;`). The model reads the input word-level code sequence from left to right in an auto-regressive manner, and makes a prediction at each position: at the end of each `print` statement, the model has to predict the correct variable value, and for all other positions, the no-output token.

Here is a short example (with `N` denoting the no-output token):

```
In:  x = 3 ; y = 7 ; x ++ ; if y < 6 : print x ; print x ;
Out: N N N N N N N N N N N N N N N N N N N N 4
```

In contrast to Fan et al. [57], we hard-code the last statement to be a `print` statement of a randomly chosen variable such that the model always has to make a prediction at the end of the sequence. The output vocabulary of the model is restricted to discrete values within a pre-determined range (here between -8 and 16), and the code sequences are constructed such that the value to be printed does not exceed this range by rejecting any statement which would result in such values. Like Fan et al. [57], we randomly generate 10,000 sequences for training and 1,000 sequences each for validation and test. With 100 statements per sequence, we obtain sequences with lengths varying from about 370 to 550, with an average length of about 450 tokens for both 3 and 5 variable cases, and for train, valid, and test sets. This code execution task requires models to maintain the values of multiple variables, which has been previously shown to be a difficult task for Transformers with only feedforward connections [57].

### B.2  Additional Results for Code Execution

**Token level print accuracy.**  First of all, as mentioned in the main text, the test accuracies reported in Table 2 are on the sequence-level, i.e., an output sequence is counted as correct only if all output

tokens in the sequence match the ground truth. The sequence level accuracy is a good evaluation measure here since for most positions in the sequence (except at the end of `print` statement) the correct target is the no-output token. This results in 0% accuracy for the Linear Transformer, which might be shocking at first glance at Table 2. Thus, we also provide the token accuracies following the `print` statements. The results can be found in Table 5. There we can see that the accuracies for the Linear Transformer are not zero: above 20% in both 3 and 5 variable cases. Nevertheless, they clearly underperform other models.

Table 5: Token-level validation accuracies (%) for the **print statements** on **code execution**. Means and stds are computed with three seeds for 3-variable and six seeds for 5-variable cases.

|  | # Variables | |
|---|---|---|
|  | 3 | 5 |
| LSTM | **99.9** $\pm$ 0.0 | **99.6** $\pm$ 0.4 |
| Transformer | 98.6 $\pm$ 0.2 | 75.5 $\pm$ 31.0 |
| Linear Transformer | 24.6 $\pm$ 0.6 | 20.7 $\pm$ 1.4 |
| Delta Net | 99.5 $\pm$ 0.1 | 97.2 $\pm$ 2.0 |
| Delta RNN | 99.5 $\pm$ 0.0 | **99.3** $\pm$ 0.2 |
| RDN | **99.6** $\pm$ 0.1 | 98.6 $\pm$ 1.4 |

**Model configurations.** The Transformer architecture in Table 2 is adopted from Fan et al. [57]: 4 layers with a hidden dimension of 256 (where we use 16 heads instead of 4) and a feedforward dimension of 1024, which yields 3.2 M parameters (like for Fan et al. [57]). We use a dropout rate of 0.1. The regular Transformer makes use of sinusoidal positional encoding (as is likely the case for Fan et al. [57]) while all other models in Table 2 don't [80, 23]. All Transformer models use pre-activation residual connections [52] and layer norm [54]. Our LSTM model in Table 2 has one LSTM layer with a dimension of 256 and an input embedding of 128 which results in 405 K parameters. We train all models with a batch size of 64 using the Adam optimiser with a learning rate of 3e-4 for Transformer-family models and a learning rate of 3e-3 for the LSTM. We clip the gradients in the LSTM model at 0.1. To train the regular Transformers, gradient accumulation was necessary to achieve the same batch size without hitting the GPU memory limit. This was not the case for space efficient linear Transformer variants. All models are trained for 200 epochs which takes no more than 23 hours for any model on a single 16 GB P100 GPU.

**Model architecture ablation.** Here we conduct a few additional experiments to understand the models' sensitivity to hyper-parameters. We restrict our analysis to the setting with 5 variables in which the performance gap between models is large (Table 2). We train deeper but thinner models with 8 layers: each with a hidden size of 128 using 8 heads and a feed-forward dimension of 256. This yields a total of 1.1 M parameters for all Transformer models except for the Delta RNN which has 1.3 M parameters. These deeper but thinner models can be trained within 10 hours using a single 16 GB V100 GPU. We present the results in the bottom part of Table 6. We don't report the performance of the regular Transformer since the 8-layer variant learns very slowly and does not improve over the initial 0% sequence-level accuracy within 200 epochs of training after which we report the performance for all models[4].

First, we observe that the Delta RNN with 8 layers can now match the performance of the baseline LSTM with 256 nodes. However, increasing the LSTM hidden size to 512 (which gives a parameter count of 1.3 M; equal to the Delta RNN's) further improves the LSTM. Second, the Delta Net still remains unstable. We tried several tricks to stabilise Transformers on algorithmic tasks [81], e.g. embedding initialisation and scaling, but with little success. The problem seems intrinsically difficult for Transformer models, though we note that one of six runs achieved a very good performance of 97.3%. Finally, we observe that the Recurrent Delta Net becomes more stable and performs better with a deep architecture.

---

[4]Extra experiments with this 8-layer regular Transformer show that after 800 epochs with a dropout rate of 0.3, a test accuracy of $89.1 \pm 2.2\%$ is achieved. This is still worse than the performance of Delta RNN trained for 200 epochs, although the comparison is not even fair due to the longer training and extra tuning.

Table 6: Test accuracies (%) on **code execution** with 5 variables. Mean, standard deviation (std), the lowest (min) and highest (max) accuracies are computed over six runs. The number of parameters (Prms.) is given in millions.

| | width | depth | mean ± std | min | max | Prms. |
|---|---|---|---|---|---|---|
| LSTM | 256 | 1 | 93.2 ± 6.1 | 84.7 | 98.5 | 0.4 |
| | 512 | | **97.7** ± 1.1 | 96.1 | 98.7 | 1.3 |
| Delta Net | 256 | 4 | 61.4 ± 20.0 | 26.2 | 85.7 | 3.2 |
| Delta RNN | | | **85.1** ± 1.9 | 83.1 | 88.6 | 3.7 |
| RDN | | | 76.3 ± 17.6 | 40.2 | 92.5 | 3.2 |
| Delta Net | 128 | 8 | 62.7 ± 36.3 | 0.1 | 97.3 | 1.1 |
| Delta RNN | | | **94.1** ± 2.7 | 88.0 | 95.8 | 1.3 |
| RDN | | | 85.0 ± 3.8 | 78.9 | 89.0 | 1.1 |

## B.3 Task Details for Sequential ListOps

The ListOps task [30] consists of list operation execution which is a typical test for hierarchical structure learning. A list is constructed using elementary list-operations written in prefix notation (typically one of six operations: maximum, minimum, median followed by floor operation, sum modulo 10, first and last element retrieval) with a random number of random arguments chosen to be either a single digit integer or a sub-list which itself has random arguments. While early research comparing self-attention to RNNs [58] has shown some advantages of recurrence in hierarchical structure learning, more recent work [59] reports Transformers to also outperform LSTMs on ListOps. Also relevant here, Tay et al. [22] report linear Transformer variants (Linear Transformers and Performers) to underperform other Transformer variants by a large margin on ListOps. It is thus natural to evaluate our models on this task as models at the intersection of recurrent and self-attention based models. We construct a simple variant of ListOps which only makes use of maximum `MAX`, minimum `MIN`, and first element retrieval `FIRST` operations. This turns out to be hard enough to shed light on the differences between our models. By construction, the targets are single digit integers. The number of arguments in each list or sub-list is random but less than the pre-determined maximum number (here set to five, following Nangia and Bowman [30]) and we control the difficulty of the task by changing the problem depth. Here is a depth-two example:

```
In:   [MAX 6 1 [FIRST 2 3 ] 0 [MIN 4 7 1] ]
Out:                                      6
```

In our setting, the task with depth 10 only contains sequences with depth 10[5]. Here, we refer to the task as "sequential ListOps", as we let the model read the sequence only once from left to right in an auto-regressive fashion. As for the code execution experiments, we randomly generate 10,000 sequences for training and 1,000 sequences each for validation and test. The lengths for the depth 10 case vary from 37 to 364 with an average length of 98 tokens. For the depth 15 case, the lengths are between 61 and 676, with an average of about 190 tokens. All experiments were conducted using a single 16 GB P100 GPU. We use the same experimental settings as in the code execution task, and the experiments for depth 10 and 15 take less than 4 and 16 hours, respectively.

## B.4 Ablation Study for the LSTM on Sequential ListOps

While the main goal of Table 2 (Sec. 4.3) was to compare different fast weight programmer variants under the same model configurations, we also pointed out that the performance of the baseline LSTM dramatically drops for the sequential ListOps task by increasing the list depth from 10 to 15. In Sec. 4.3, we hypothesised the reason for the performance drop of the LSTM for the depth-15 case of sequential ListOps to be the small hidden size of the LSTM and the increase of sequence lengths in

---

[5]However, here the *depth* is simply defined as the depth of nested operations. Since the used operations do not always have to evaluate all arguments to obtain the result, the *effective computation* may be shallower. This problem has been addressed in a better version of ListOps in our more recent work [82].

the depth-15 case. Here we provide the corresponding ablation study. Table 7 shows the performance of the LSTM with different hidden layer sizes. We find that increasing the hidden size effectively help the LSTM on this task.

Table 7: Test accuracies (%) with standard deviations over three runs for the **LSTM** on the **depth-15** case of **Sequential ListOps**.

| Hidden size | Mean accuracy ± std |
|---|---|
| 256 | 24.4 ± 1.1 |
| 1,024 | 24.4 ± 0.7 |
| 2,048 | 35.9 ± 13.0 |
| 4,096 | **72.2** ± 1.6 |

## C   Experimental Details and Additional Results for RL in Atari 2600

**Settings.**   We use the `polybeast` implementation from `Torchbeast` [67] with modifications limited to model architectures. We train all our models using RMSProp [83] with a learning rate of 0.0006, an epsilon of 0.01, and gradient clipping at 40. We use entropy regularisation with a weight of 0.01. The backpropagation span is 50 and the batch size is 32. The model architecture and evaluation method is described in the main text. All Transformer variants make use of pre-activation residual connections [52, 12] and layer norm [54]. The number of actors for IMPALA is 48. No action repeat is used. No time limit is set for evaluation. Rewards are clipped between -1 and 1. The OpenAI Gym implementation of the Atari learning environment [84] is used. The only source of stochasticity is the default sticky action. We train expert models using the game specific action spaces (models for *Amidar* and *James Bond* were trained with an action space size of 6, which is smaller than the full action space but enough to play these games). We train on 2 GPUs (either 16 GB P100 or 32 GB V100). An experiment for one game takes about 1.5 days. Evaluation is done at 50 M and 200 M environmental steps, which are reported in Table 9 and 10. For cases where performance did not improve after 50 M and 200 M, we report the performance at 50 M again in Table 10 (we experienced this for *Bank Heist* and *Robotank*; for *Pong* 50 M steps are enough to consistently achieve the perfect score).

In what follows, we provide additional model comparisons.

**Feedforward vs. LSTM.**   On Atari, models without recurrence are also known to perform well in many environments [68]. Since it is not easy to compare RL systems across different settings [85], we train our own feedforward baseline. The feedforward baseline is simply obtained by removing the LSTM layer in the LSTM model, which corresponds to the model of Espeholt et al. [65]. At 50 M steps (Figure 5; *orange*), there are 8 games in which the feedforward baseline clearly outperforms the LSTM, and in 8 other games the trend is reversed. At 200 M steps (Figure 6; *orange*), the LSTM performs clearly better in 10 games, whereas the feedforward net clearly dominates only in 4 games.

**Feedforward baseline with more parameters.**   In the comparison above, the LSTM baseline has 1.6 M parameters, more than the 1.1 M parameters of the feedforward baseline (while we note that the RDN has slightly fewer parameters than the LSTM, namely, 1.5 M). To verify that the improvements obtained by the LSTM are not due to the increased parameter count, we build a larger feedforward baseline with 1.7 M parameters by replacing the LSTM layer by one feedforward highway-gated layer [53] (to keep it as similar as possible to the LSTM baseline). Here the output from the vision stem is first projected to a 320-dimensional vector which is followed by a 256-dimensional highway-gated layer. We evaluate this model on four environments on which the LSTM outperforms the 1.1 M-param feedforward baseline. Table 8 shows the corresponding results. The extra parameters yield improvements only on *S. Invader*, without matching LSTM's performance. So we can confirm that the dominance of LSTM over feedforward models in these games is not simply due to the higher parameter count.

**Recurrent Delta Net vs. Delta Net.**   We also compare the Recurrent Delta Net to a stronger baseline, the Delta Net. The results are shown in Figures 7 and 8 (*sky blue*). While the RDN performs equally well or better than the baseline Delta Net on 13 games at 200 M steps, there are also 7 games

Table 8: Performance of feedforward baseline with more parameters.

| | Params. | Berzerk | Gopher | Seaquest | S. Invader |
|---|---|---|---|---|---|
| LSTM | 1.5 M | 1,150 ± 92 | 124,914 ± 22,422 | 12,643 ± 1,627 | 137,657 ± 2,276 |
| FF | 1.1 M | **343** ± 23 | **61,350** ± 3,891 | **667** ± 1 | 53,455 ± 6,694 |
| FF gated | 1.7 M | 320 ± 29 | 42,851 ± 7,653 | 660 ± 0 | **95,629** ± 11,991 |

where the Delta Net is better. We thus can not guarantee strict benefits of additional recurrence here. Again we note that compared to other models, both the Delta Net and Recurrent Delta Net achieve outstanding performance on *Q*Bert*.

**Delta RNN vs. LSTM.** We also evaluate the Delta RNN (Sec. 3.1) in this RL setting. We first compare it to the LSTM baseline. As shown in Figures 9 and 10 (*green*), the Delta RNN clearly outperforms the LSTM on a few games at 50 M steps. However, the performance gaps reduce across all games after 200 M steps. Overall, the performance is close in 7 games, in favour of the LSTM in 8 games, and in favour of the Delta RNN in 5 games.

**Recurrent Delta Net vs. Delta RNN.** Finally, we also compare the Recurrent Delta Net to the Delta RNN. Figures 11 and 12 (*grey*) present our results. In 16 games, the relative performance gap is within 50%. In one game (*Seaquest*), the Delta RNN outperforms the RDN. In 3 games, the RDN clearly outperforms the Delta RNN at 200 M steps.

Overall, the Recurrent Delta Net tends to yield decent performance compared to all baselines. While the performance gaps between the Recurrent Delta Net and the Delta RNN are rather close, the Recurrent Delta Net performs particularly well in a few games. As mentioned in the main text, trying deeper architectures might be a straight-forward way to obtain better scores.

## D Comments on Nomenclature

To simplify references to specific Fast Weight Programmers, we gave short names to all of them, such as Delta RNN or Recurrent Delta Net. We did not cover, however, many other possible combinations of slow and fast networks as well as update rules (which are the elementary programming instructions of FWPs). This calls for a systematic nomenclature to specify the various FWP types. For a given FWP, one could use "*slow-net/update-rule*" as a *prefix* and "*fast-net*" architecture as a *suffix*. For example, the Delta RNN is an FWP with a fast RNN and a feedforward slow net using the delta rule as elementary programming instruction. Therefore, using the convention above, the full name of the Delta RNN would be "*Feedforward/Delta fast RNN*." The full name of the Recurrent Delta Net would be "*Recurrent/Delta fast Linear Net*," and so on. This is also compatible with the baseline Delta Net, whose full name would be "*Feedforward/Delta fast Linear Net*."

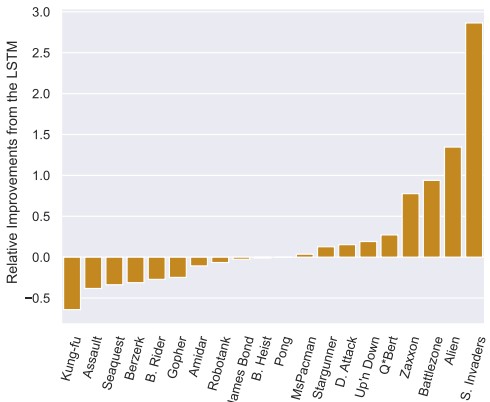

Figure 5: Rel. improvements in test scores obtained by the **feedforward baseline** compared to LSTM after **50 M** env. steps.

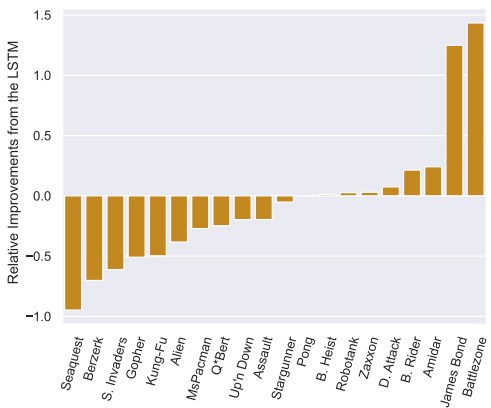

Figure 6: Rel. improvements in test scores obtained by the **feedforward baseline** compared to LSTM after **200 M** env. steps.

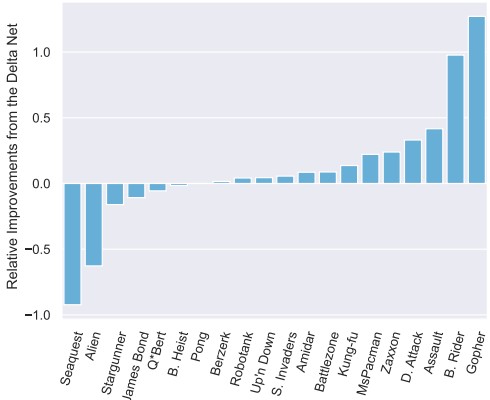

Figure 7: Rel. improvements in test scores obtained by the **Recurrent Delta Net** compared to the **Delta Net** after **50 M** env. steps.

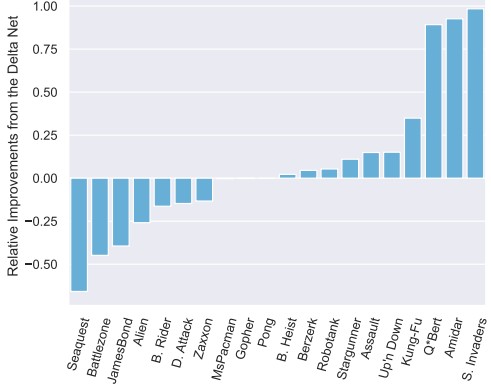

Figure 8: Rel. improvements in test scores obtained by the **Recurrent Delta Net** compared to the **Delta Net** after **200 M** env. steps.

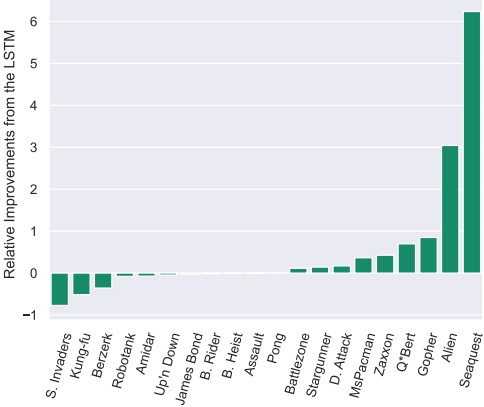

Figure 9: Rel. improvements in test scores obtained by the **Delta RNN** compared to the **LSTM** after **50 M** env. steps.

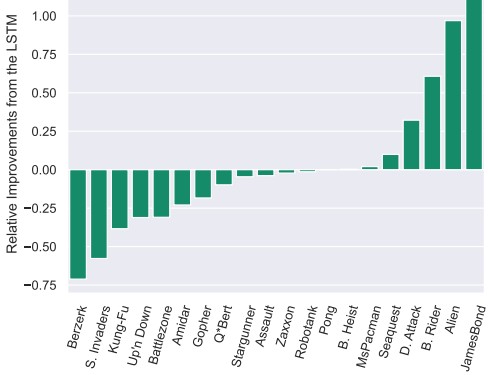

Figure 10: Rel. improvements in test scores obtained by the **Delta RNN** compared to the **LSTM** after **200 M** env. steps.

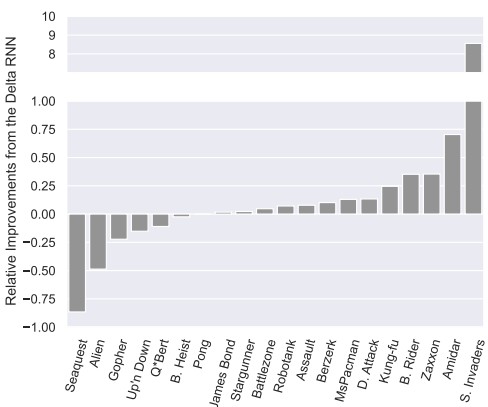
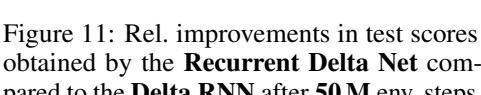

Figure 11: Rel. improvements in test scores obtained by the **Recurrent Delta Net** compared to the **Delta RNN** after **50 M** env. steps.

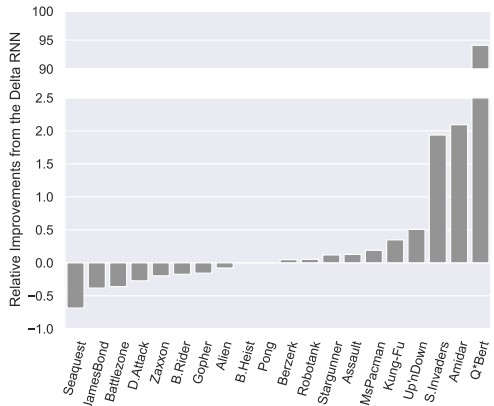

Figure 12: Rel. improvements in test scores obtained by the **Recurrent Delta Net** compared to the **Delta RNN** after **200 M** env. steps.

Table 9: Performance after **50 M** environmental steps. Reported scores are mean and std of 5 mean-scores obtained over 30 episodes (total of 150 different test episodes). We remind the reader that we denote the Linear Transformer [19] as LT, and our Recurrent Delta Network as RDN. The numbers of parameters are: 1.1 M for the feedforward model, 1.6 M for the LSTM, 1.5 M for the Linear Transformer, the Delta Net, and the Recurrent Delta Net, and finally 1.6 M for the Delta RNN.

|  | Feedforward | LSTM | LT | Delta Net | RDN | Delta RNN |
|---|---|---|---|---|---|---|
| Alien | $1,985 \pm 90$ | $846 \pm 81$ | $2,135 \pm 184$ | $\mathbf{4,704} \pm 452$ | $1,754 \pm 48$ | $3,420 \pm 834$ |
| Amidar | $208 \pm 11$ | $233 \pm 10$ | $320 \pm 16$ | $339 \pm 28$ | $\mathbf{368} \pm 23$ | $216 \pm 14$ |
| Assault | $4,658 \pm 2,147$ | $7,551 \pm 1,774$ | $2,764 \pm 380$ | $5,710 \pm 2,643$ | $\mathbf{8,088} \pm 2,851$ | $7,503 \pm 2,794$ |
| Battlezone | $\mathbf{12,267} \pm 620$ | $6,327 \pm 380$ | $933 \pm 351$ | $6,780 \pm 461$ | $7,373 \pm 431$ | $7,040 \pm 1,098$ |
| Berzerk | $326 \pm 21$ | $\mathbf{474} \pm 17$ | $323 \pm 6$ | $331 \pm 24$ | $336 \pm 27$ | $305 \pm 8$ |
| B. Heist | $323 \pm 13$ | $\mathbf{327} \pm 11$ | $309 \pm 11$ | $321 \pm 8$ | $316 \pm 10$ | $324 \pm 10$ |
| B. Rider | $9,932 \pm 1,592$ | $13,638 \pm 1,571$ | $6,695 \pm 941$ | $9,185 \pm 630$ | $\mathbf{18,156} \pm 1,522$ | $13,429 \pm 884$ |
| D. Attack | $36,255 \pm 3,566$ | $31,447 \pm 1,850$ | $8,939 \pm 950$ | $31,359 \pm 3,362$ | $\mathbf{41,726} \pm 6,308$ | $36,807 \pm 3,700$ |
| Gopher | $10,356 \pm 378$ | $13,765 \pm 808$ | $8,197 \pm 1,720$ | $8,707 \pm 2,381$ | $19,775 \pm 1,448$ | $\mathbf{25,445} \pm 1,963$ |
| James Bond | $2,942 \pm 56$ | $3,020 \pm 68$ | $2,425 \pm 174$ | $\mathbf{3,338} \pm 137$ | $2,979 \pm 176$ | $2,929.3 \pm 408$ |
| Kung-fu | $5,449 \pm 82$ | $\mathbf{15,216} \pm 818$ | $3,722 \pm 330$ | $8,095 \pm 240$ | $9,201 \pm 384$ | $7,388 \pm 491$ |
| MsPacman | $1,737 \pm 53$ | $1,676 \pm 86$ | $1,647 \pm 101$ | $2,116 \pm 30$ | $\mathbf{2,584} \pm 121$ | $2,287 \pm 32$ |
| Pong | $21 \pm 0$ | $21 \pm 0$ | $21 \pm 0$ | $21 \pm 0$ | $21 \pm 0$ | $21 \pm 0$ |
| Q*Bert | $4,967 \pm 266$ | $3,905 \pm 252$ | $4,693 \pm 195$ | $6,248 \pm 204$ | $5,897 \pm 357$ | $\mathbf{6,626} \pm 240$ |
| Robotank | $7.1 \pm 0.7$ | $\mathbf{7.6} \pm 0.7$ | $4.8 \pm 0.3$ | $7.2 \pm 0.7$ | $7.5 \pm 0.8$ | $7.0 \pm 0.5$ |
| Seaquest | $469 \pm 1$ | $708 \pm 1$ | $1,812 \pm 61$ | $\mathbf{8,853} \pm 937$ | $686 \pm 1$ | $5,123 \pm 335$ |
| S. Invaders | $\mathbf{48,150} \pm 7,233$ | $12,461 \pm 1,624$ | $2,345 \pm 74$ | $25,769 \pm 10,156$ | $27,213 \pm 3,359$ | $2,847 \pm 10$ |
| Stargunner | $9,397 \pm 2,193$ | $8,337 \pm 1,094$ | $8,915 \pm 713$ | $\mathbf{11,599} \pm 3,454$ | $9,737 \pm 1,396$ | $9,523 \pm 2,214$ |
| Up'n down | $\mathbf{185,632} \pm 16,490$ | $155,847 \pm 15,318$ | $57,435 \pm 2,283$ | $120,806 \pm 16,261$ | $126,140 \pm 19,078$ | $148,759 \pm 28,492$ |
| Zaxxon | $4863 \pm 872$ | $2,737 \pm 121$ | $2,719 \pm 701$ | $4,265 \pm 263$ | $\mathbf{5,285} \pm 504$ | $3,903 \pm 648$ |

Table 10: Performance after **200 M** environment steps. Reported scores are mean and std of 5 mean-scores obtained over 30 episodes (total of 150 different test episodes). We remind the reader that we denote the Linear Transformer [19] as LT, and our Recurrent Delta Network as RDN. In cases where performance did not improve after 50 M, we report the performance at 50 M.

| | Feedforward | LSTM | LT | Delta Net | RDN | Delta RNN |
|---|---|---|---|---|---|---|
| Alien | $3,816 \pm 139$ | $6,184 \pm 558$ | $4,751 \pm 530$ | $\mathbf{15,133} \pm 1,122$ | $11,220 \pm 621$ | $12,177 \pm 968$ |
| Amidar | $433 \pm 27$ | $349 \pm 22$ | $646 \pm 32$ | $432 \pm 27$ | $\mathbf{832} \pm 11$ | $269 \pm 17$ |
| Assault | $6,407 \pm 3,430$ | $7,977 \pm 2,611$ | $6,465 \pm 1,437$ | $7,525 \pm 1,703$ | $\mathbf{8,647} \pm 3,061$ | $7,670 \pm 952$ |
| Battlezone | $60,527 \pm 12,345$ | $\mathbf{24,873} \pm 1,240$ | $2,667 \pm 386$ | $19,907 \pm 1,409$ | $10,980 \pm 1,104$ | $17,180 \pm 1,493$ |
| Berzerk | $343 \pm 23$ | $\mathbf{1,150} \pm 92$ | $480 \pm 38$ | $333 \pm 7$ | $348 \pm 17$ | $332 \pm 17$ |
| B. Heist | $\mathbf{331} \pm 10$ | $327 \pm 11$ | $317 \pm 8$ | $321 \pm 8$ | $328 \pm 10$ | $328 \pm 8$ |
| B. Rider | $21,873 \pm 2,000$ | $18,024 \pm 933$ | $22,444 \pm 755$ | $28,594 \pm 5,508$ | $23,934 \pm 2,292$ | $\mathbf{28,973} \pm 3,663$ |
| D. Attack | $74,904 \pm 10,941$ | $69,750 \pm 9,593$ | $57,715 \pm 5,009$ | $78,601 \pm 16,907$ | $67,039 \pm 5,714$ | $\mathbf{92,205} \pm 17,933$ |
| Gopher | $61,350 \pm 3,891$ | $\mathbf{124,914} \pm 22,422$ | $48,261 \pm 7,727$ | $86,168 \pm 5,069$ | $86,008 \pm 11,815$ | $101,974 \pm 10,200$ |
| James Bond | $\mathbf{56,459} \pm 7,292$ | $25,106 \pm 5,889$ | $16,223 \pm 1,118$ | $54,336 \pm 7,165$ | $32,923 \pm 7,968$ | $53,344 \pm 4,768$ |
| Kung-fu | $12,292 \pm 613$ | $\mathbf{24,447} \pm 407$ | $13,969 \pm 803$ | $15,064 \pm 929$ | $20,319 \pm 363$ | $15,068 \pm 513$ |
| MsPacman | $2,499 \pm 141$ | $3,431 \pm 197$ | $3,052 \pm 128$ | $\mathbf{4,180} \pm 139$ | $4,168 \pm 585$ | $3,500 \pm 205$ |
| Pong | $21 \pm 0$ | $21 \pm 0$ | $21 \pm 0$ | $21 \pm 0$ | $21 \pm 0$ | $21 \pm 0$ |
| Q*Bert | $8,655 \pm 371$ | $11,513 \pm 910$ | $8,389 \pm 349$ | $521,839 \pm 36,192$ | $\mathbf{987,275} \pm 0$ | $10,381 \pm 1,259$ |
| Robotank | $7.8 \pm 0.8$ | $7.6 \pm 0.7$ | $7.7 \pm 0.9$ | $7.5 \pm 0.4$ | $\mathbf{7.9} \pm 0.6$ | $7.5 \pm 0.5$ |
| Seaquest | $667 \pm 1$ | $12,643 \pm 1,627$ | $12,425 \pm 1,910$ | $12,790 \pm 1,512$ | $4,373 \pm 504$ | $\mathbf{13,898} \pm 1,674$ |
| S. Invaders | $53,455 \pm 6,694$ | $137,657 \pm 2,276$ | $2,333 \pm 110$ | $86,132 \pm 5,483$ | $\mathbf{170,871} \pm 80$ | $58,181 \pm 14,987$ |
| Stargunner | $11,564 \pm 4,598$ | $12,194 \pm 7,038$ | $12,035 \pm 6,995$ | $11,734 \pm 6,827$ | $\mathbf{13,026} \pm 6,431$ | $11,635 \pm 6,065$ |
| Up'n down | $185,632 \pm 16,490$ | $231,157 \pm 10,603$ | $\mathbf{252,555} \pm 16,331$ | $208,563 \pm 22,803$ | $240,003 \pm 26,849$ | $159,296 \pm 25,013$ |
| Zaxxon | $\mathbf{11,960} \pm 538$ | $11,619 \pm 663$ | $7,371 \pm 932$ | $1,0523 \pm 568$ | $9,126 \pm 313$ | $11,365 \pm 678$ |