# OpenReview forum: "Going Beyond Linear Transformers with Recurrent Fast Weight Programmers"
_NeurIPS.cc/2021/Conference — NeurIPS 2021 Poster_

### Official Review · Reviewer_piWh · 2021-07-02

**Rating:** 7
**Confidence:** 4

**Summary:**

This paper studies linear attention variants within transformers from the perspective of fast weight programmers from the 90s.  In particular, it explores new more sophisticated variants that have recurrence on both the slow (fixed after training) and fast networks (generated) as opposed to previous variants such as Delta Net that uses one-layer feedforward networks for both.  The evaluation shows that recurrence leads to improved results compared to both standard and linear transformers.

**Limitations And Societal Impact:**

The paper discusses the limitations of the proposed models and explores their speed-quality tradeoffs in the evaluation section. I would suggest that the authors discuss the applicability of the proposed variants on non-autoregressive tasks and whether the proposed methods can handle them (and why not if they can't) so that the scope of this paper is clear to the readers. There is also a section dedicated to the potential negative societal impact.

**Main Review:**

The exploration of more sophisticated fast weight programmers for linear attention is an interesting and valuable direction since it enhances the understanding of the linear attention variants. Even though the connection to fast weight programmers itself is not new the proposed variants have sufficient novelty and increase the expressiveness in maintaining the internal state when used for attending to long sequences.

The technical exposition was quite clear and well-positioned with respect to prior work. The motivation for using recurrence becomes apparent given the simplicity of the networks used in previous variants. The study reveals that using a Delta Net with simple recurrence leads to best results and more sophisticated recurrence based on LSTMs removes the efficiency benefits in linear transformers. One interesting insight from this finding is that it is not possible to increase the expressiveness of linear attention beyond a certain point without being slower compared to the original transformer; albeit the model would still be scalable while the original transformer is not.

The evaluation is thorough, contains controlled comparisons, and uses a configuration similar configurations to prior work. The results support the claims in the paper and shed light on the strengths of different variants. I also found the details provided in the supplementary quite helpful. Perhaps one limitation is that the evaluation does not include any non-autoregressive tasks.

Questions:
- Is the difference in LSTM performance on the code execution task from prior work due to the data creation process or some other factor?

- Can the proposed attention variants be applied to non-autoregressive tasks? Recent efficient variants are applicable to such settings, including linear transformers, Performer/RFA.

- The statement that fast weight variants maintain the standard one-dimensional state and two-dimensional fast weight states in Sec 3.2 was a bit confusing. Where do these dimension sizes come from?


**Time Spent Reviewing:**

3.5

---

> ### Author Response · Authors · 2021-08-09
> **Response to Reviewer piWh**
>
> We thank the reviewer for many positive comments and valuable feedback.
> We are pleased to hear that the reviewer found the supplemental material useful. Please find our responses to the questions below:
>
> > *Is the difference in LSTM performance on the code execution task from prior work due to the data creation process or some other factor?*
>
> Our best guess is that the difference comes from the hyper-parameter tuning of the baselines. It should not be surprising for practitioners that the LSTM is capable of maintaining the state of a few symbols. In fact, we even chose this task based on our expectation that the LSTM would be a strong baseline.
>
> The poor performance of the LSTM reported in the previous work (Fan et al. [41]) had indeed surprised one of the reviewers of said work (see AnonReviewer2; we are not related to this review) https://openreview.net/forum?id=OCm0rwa1lx1.
>
> > *Can the proposed attention variants be applied to non-autoregressive tasks? Recent efficient variants are applicable to such settings, including linear transformers, Performer/RFA.*
>
> No, our models are only relevant for auto-regressive scenarios, just like other recurrent neural networks such as the LSTM. If a bi-directional context is needed, we have no other choice but to use two separate RNNs and run them in both directions, again just like with the LSTM.
> As suggested by the reviewer, we will add a paragraph to clarify this in a revised version. Thank you for pointing this out.
>
> > *The statement that fast weight variants maintain the standard one-dimensional state and two-dimensional fast weight states in Sec 3.2 was a bit confusing. Where do these dimension sizes come from?*
>
> We agree with the reviewer that this formulation was confusing. Our original intention was to note that we can categorize our models as a memory augmented recurrent NN with one-dimensional RECURRENT state (an RNN state vector) and a 2-dimensional memory (a matrix of fast weights). But indeed, citations (e.g. Graves' Neural Turing Machine) were also missing there. As this was also pointed out by another reviewer, we will edit this sentence in the revised version. We thank the reviewer for pointing this out.

---

> > ### Comment · Reviewer_piWh · 2021-08-11
> > **Response to rebuttal**
> >
> > Thank you for answering my questions; it would be useful if they are reflected in the final paper. I appreciate the thoughtful replies and clarifications to the concerns raised by all the reviewers. Having read the other discussions, I am still positive about the paper but I would like to see what the other reviewers think.

---

### Official Review · Reviewer_7TUg · 2021-07-16

**Rating:** 6
**Confidence:** 4

**Summary:**

The paper presents a new series of transformer architectures with novel linear attention mechanisms, all based on the formalism of fast weight programmers (FWPs). The fast weight programming perspective views attention as a "slow" network producing internal weight updates to a "fast" weight hyper-network, where an update to the fast weight matrix is predicted and accumulated at each time step in the input sequence. Therefore the fast weight network changes it's parameters during processing of a single sequence, while the slow network is updated using standard gradient descent on the loss function across sequences. Using this perspective of fast weights, the paper designs several novel attention replacements: (1) recurrent architectures for the fast weight network, (2) deep feedforward networks, and (3) a nested recurrent architecture, the Recurrent Delta Net (RDN), where both the fast and slow network include a recurrence. Results on language modeling, code execution and reinforcement learning domains reveal the benefits of approaching linearized attention under this formalism, with the RDN in particular showing advantages over even standard softmax-attention transformers.

**Limitations And Societal Impact:**

The authors have discussed the limitations of their model. The model also does not seem to have any significant potential for negative societal impact.

**Main Review:**

The paper is well-written and clear to follow. The formalism of fast weight programmers makes understanding the attention variants intuitive, and the design choices of the network architectures presented using this perspective follow naturally.  The experimental results seem to support the viability of the architectures, with similar or better results to both standard transformer and linear transformer baselines. Adding recurrence to the fast weight mechanism seems to be especially important on the code execution and list ops domains, which largely close the gap with the LSTM baseline.

While the paper presents a formalism from which a few novel architectures are designed, there is a downside that the results do not always provide a clear advantage to one particular architecture over another. For example, Delta LSTM does best out of the presented architectures at language modeling, but is twice as slow as a regular transformer and still underperforms it at test perplexity. As another example, at code execution it is either the RDN or the Delta RNN which are best performing. This somewhat limits the significance of the results, since a practitioner can't immediately choose one architecture and might need to ablate over many of them. It could make the narrative of the experiments section cleaner if the authors chose to focus on a particular architecture, and demonstrated a clearer performance improvement under some chosen metrics (wall-clock training time, inference time, etc.).

It would also be interesting to see how these models perform at larger scale. I would be very interested to see the scaling properties of the recurrent attention architectures, although I understand that running these experiments could be expensive.

Additionally, is there an intuition on why any of the memory architectures on Atari would be particularly helpful? The results seem to support that, but a more thorough understanding of why exactly would be informative. Is this caused by increased parameter count, ease-of-optimization, or is the memory actually being used effectively in the policy?

In conclusion, I believe the paper has presented an interesting perspective on recurrent attention and, despite some issues with the focus of the results, I vote for acceptance.

**Time Spent Reviewing:**

4

---

> ### Author Response · Authors · 2021-08-09
> **Response to Reviewer 7TUg**
>
> We thank the reviewer for many positive comments and valuable feedback.
>
> > *... there is a downside that the results do not always provide a clear advantage to one particular architecture over another. ... It could make the narrative of the experiments section cleaner if the authors chose to focus on a particular architecture.*
>
> We fully agree with the reviewer that, as we do not focus on studying a single new model, we do not have a clear winner architecture in our study. Indeed, it could have been even easier for us if we had focused our study e.g. solely on the Delta RNN and completely omitted the RDN. However, we made a decision to conduct a full investigation on both the slow and fast net sides of the story and to present the whole as a new exploratory work for the linear Transformers (a bit in the flavor of exploration works on recurrent neural network architectures around 2015 such as Jozefowicz et al. ICML 2015 [1’]). And we found both RDN and Delta RNN to be interesting variants of linear Transformers. In this exploration, we also obtain a clear trend that the original linear Transformer underperforms other fast weight variants across all four tasks we investigated (we note that we also evaluated the vanilla linear Transformer in the RL setting in the supplemental material Sec. C, Figure 6; we should have stressed more on this in the main text) while the result of Schlag et al. [18] was limited for one language modelling task. We believe this to be an important confirmation.
>
>
> > *It would also be interesting to see how these models perform at larger scale. I would be very interested to see the scaling properties of the recurrent attention architectures, although I understand that running these experiments could be expensive.*
>
> Yes, with a limited compute resource, we had to make a choice of evaluating our models on multiple task domains, instead of training a few big models on a single task (e.g. language modelling). However, at least for the RL experiment, we show benefits of scaling up our RDN models (Table 4; Line 322-327) and effectively obtain substantial improvements by simply increasing the number of layers and the hidden layer dimension.
>
> > *Additionally, is there an intuition on why any of the memory architectures on Atari would be particularly helpful? The results seem to support that, but a more thorough understanding of why exactly would be informative. Is this caused by increased parameter count, ease-of-optimization, or is the memory actually being used effectively in the policy?*
>
> While most games here should in principle benefit from memory as a partially observable environment,
> we can only provide an empirical answer to the question whether a feedforward model with a limited context window (e.g. four frames) and with an advantage to be easier to optimise can outperform some other models with proper memory.
> It is known from previous works (e.g. Mott et al. [52]) that feedforward models with a small time window (e.g. 4 frames) can already perform well on multiple Atari games, while on other games the LSTM clearly outperforms it.
> This is something we also confirmed ourselves while comparing the feedforward baseline to the LSTM baseline (Sec. C in the supplemental material, from Line 728-735 and Figure 4).
>
> Regarding the parameter count, it is true that the LSTM baseline with 1.6 M parameters has more parameters than the feedforward baseline with 1.1 M (while we note that the RDN has slightly less parameters than the LSTM, 1.5 M). In order to answer the reviewer’s question, we ran some extra experiments using a larger feedforward baseline with 1.7 M parameters. We built this larger model by replacing the LSTM layer in the LSTM model by one large feedforward highway-gated layer (to keep it as similar as possible to the LSTM baseline but without memory).
>
> We evaluated this 1.7 M-param feedforward model on four environments on which the LSTM outperforms the 1.1 M-param feedforward baseline.
> We obtain the following scores:
>
> Env | new FF (1.7 M param) |  FF (1.1 M param) | LSTM (1.5 M param)
>
> Berzerk | 320 +- 29 | 343 +- 23 | 1,150 +- 92
>
> Gopher | 42,851 +- 7,653 | 61,350 +- 3,891 | 124,914 +- 22,422
>
> Seaquest | 660 +- 0 |  667 +- 1 | 12,643 +- 1,627
>
> S. Invader | 95,629 +- 11,991 | 53,455 +- 6,694 | 137,657 +- 2,276
>
> The extra parameters thus only gave improvements on "S. Invader"
> which are not enough to match the LSTM's performance.
> So we can confirm that the dominance of the LSTM on these games over the feed-forward model is not due to the higher parameter counts. We thank the reviewer for bringing this to our attention. We will add this results in the appendix of a revised version.
>
> If you think our responses strengthened your assessment of our paper and/or you think our paper should be accepted, we would really appreciate it a lot if you can consider increasing the score. Thank you.
>
> [1’] Jozefowicz et al. An empirical exploration of recurrent network architectures. ICML 2015

---

> > ### Author Response · Authors · 2021-08-31
> > **Friendly reminder**
> >
> > This is just a friendly reminder about the NeurIPS rebuttal deadline.
> > Please let us know if you have any remaining questions. Thank you!

---

### Official Review · Reviewer_u3vG · 2021-07-17

**Rating:** 6
**Confidence:** 4

**Summary:**

This work proposes modifications to the linear transformer paradigm that adds recurrent dependencies to past transformer embeddings. The resulting "Recurrent Fast Weight Programmer" model proposed is shown to outperform transformers on multiple sequence learning tasks. Furthermore, the proposed method remains competitive with RNNs in tasks where RNN architectures are still dominant over transformers, implying a best of both worlds feature in various sequence learning settings.

**Limitations And Societal Impact:**

Suggestions for improvement:

-Introduction and specifically lines 36-45 begin discussion of the architectural modifications that are hard to understand without any formulation. (lines 42-43 seem contradictory)

-Explain what are the scaling differences between Delta Delta Net and the size of the backpropagation graph if another meta-level of slow to fast weights are added?

-Lines 186-190 need a better explanation for the formulations of the slow weights. More capacity and additional memory are too vague. Why formulate the fast weights with 1-D states and weights as 2-D states?

-Since some of these models are not limited by the token span as much as vanilla transformers, why not try the models that can support it much longer sequence spans?

-Lines 216-218 like earlier lines are too vague, an intuition for why the slow network requires additional input (and also untested). It seems like the lack of strong reasoning behind these architecture modifications makes conclusions drawn from the results ambiguous.

-The surprisingly good performance of the LSTM on the code execution task is also not explained or analyzed. The lack of analysis here is unfortunate because this deviates from what is expected and is worth understanding.

-The falloff in LSTM performance at higher depth (depth 15) is also an interesting observation that should be expanded upon.

**Main Review:**

The text presents several potential variations of the linear transformer formulation which incorporate various properties such as recurrence and differing types of interactions between fast and slow weights. But the motivation for such architectural modifications is unclear from the presentation of the paper. Why is the DeltaNet formulation necessary as compared to the Linear Transformer? What aspect of the DeltaNet formulation improves model capacity?

Overall, the connections between linear transformers and Fast Weight Programmers are interesting because it may give rise to explanations of where the benefits and limitations exist among both styles of architectures. Furthermore, it may help explain what deviations could be expected in performance from the original/dominant Transformer. These differences are seen in the experiments of this work. But this work has little to no hypotheses and therefore no subsequent analysis of how these modifications improve/don't improve performance. For instance, how is the \beta gating function behaving when given various difficulty levels of the code execution and sequence list tasks?

Post Rebuttal: I have gone over the author responses and discussions from other reviewers and have decided to increase my score. My main concern is still the lack of exploration into the function of the additional modifications proposed in the RDN model. Without knowing what is systematically different between the proposed model after learning a task and other models, a never-ending paper stream of variations of linear transformers that do not necessarily perform any better than the original transformer formulation in the real world can result. But I agree with the author's responses that this may be too much to ask for in a paper submission.

**Time Spent Reviewing:**

4

---

> ### Author Response · Authors · 2021-08-09
> **Response to Reviewer u3vG, part 1/2**
>
> We thank the reviewer for valuable time reviewing our work.
> There seems to be a few misunderstandings which we'd like to resolve.
>
> > *Why is the DeltaNet formulation necessary as compared to the Linear Transformer? What aspect of the DeltaNet formulation improves model capacity?*
>
> The memory capacity problem of the original linear Transformer, as well as, the benefit of introducing the delta rule to it (thus obtaining the DeltaNet) have been studied in depth by Schlag et al. [18].
> This is not our contribution.
> We thus opted to include only the description of the DeltaNet without its justification in the main text.
>
> Here is however a short answer to the reviewer’s questions:
> Essentially, Schlag et al. [18] point out that the operations in the original linear Transformer can be seen as a fast weight/associative memory mechanism: write operation as in Eq. 6 and retrieval as in Eq 7. They demonstrate that such a memory will necessarily reach a memory capacity limit (i.e. have retrieval noises) if the number of associations (key/value pairs) to be stored exceeds the key dimension. They hypothesised that the way the original linear Transformers update its memory (Eq. 6) is suboptimal, as it only naively sums up all associations. The delta rule they propose instead implements a mechanism for explicitly updating associations by removing the old association when adding a new one, which is empirically shown to outperform the original sum update rule.
>
> > *Overall, the connections between linear transformers and Fast Weight Programmers are interesting ...  But this work has little to no hypotheses and therefore no subsequent analysis of how these modifications improve/don't improve performance. For instance, how is the beta gating function behaving when given various difficulty levels of the code execution and sequence list tasks?*
>
> This work is an empirical architecture exploration (an example of such a type of work is [1’] Jozefowicz et al. ICML 2015), and it is not really possible to conduct a hypothesis/test experimentation at the level expected by the reviewer. The example below about beta is a good illustration.
>
> The beta variable is the dynamic learning rate of the delta update rule for updating the fast weights proposed in Schlag et al. [18]. While it is intuitive to understand it from the perspective of weight update rule (it’s simply a scale for the update term), it is not straightforward to hypothesise its exact behaviour for a specific task, especially within a deep network. In fact, such is the case for many popular models in deep learning e.g. the forget gate of LSTM is intuitive, but can we hypothesise its behaviour given some specific sequence processing task? Probably not, or at least that does not seem obvious to us.
>
> Also, we again note that the delta update rule is not a contribution of this work but from Schlag et al. [18].
>
> > *Introduction and specifically lines 36-45 begin discussion of the architectural modifications that are hard to understand without any formulation. (lines 42-43 seem contradictory)*
>
> As a part of introduction, we had to keep the formulation in Lines 36-45 rather brief on the conceptual level. As we clarify in line 39, we do review the concept of FWP in detail later in the background Sec 2 for readers who are not familiar with the concept of FWP, and the entire Sec 3 is dedicated to fully describe the proposed models.
> Regarding 42-43, we are not fully sure what contradiction the reviewer is pointing at. Could you please explain?
> The corresponding lines are: “In the case of existing linear Transformer variants, the slow and fast nets are simple one layer feedforward NNs. Here we augment them with recurrent connections to obtain recurrent FWPs (RFWPs)”. But we can not find any contradiction here.
>
> > *Explain what are the scaling differences between Delta Delta Net and the size of the backpropagation graph if another meta-level of slow to fast weights are added?*
>
> We are not sure to fully understand the reviewer’s question here, so please correct us if our interpretation is wrong. We assume the actual question to be: “how does the model scale from DeltaNet to DeltaDeltaNet?"
>
> Let’s assume a DeltaNet with a dimension D for all query, key, value and input vectors.
> Then its slow weight matrix (the projection matrix) is of size “D\*(3\*D+1)” as it projects a D-dimensional input to query, key, value vectors (3\*D) and a scalar beta (+1) which are needed to maintain D\*D fast weight matrix using the delta rule.
> Now we can express the dimensionality of a DeltaDeltaNet in terms of D, whose fast network is a DeltaNet with the dimensionality above. The size of its fast weight matrix is thus D\*(3\*D+1). In order to maintain a fast weight of this dimension using the delta rule, we need key and query vectors of size D, a value vector of size (3\*D+1), and a scalar beta (+1). The slow weight matrix has to produce all these variables with a total dimension of (5\*D+2) from the input of size D. Therefore, the size of the slow weight matrix in DeltaDeltaNet is D\*(5\*D+2). The layer would have to store two fast weights: one of size D\*(3\*D+1) and another one of size D\*D.
>
> We hope this description gives a concrete sense of the computational scale involved in DeltaDeltaNet. We’d like to also emphasise that we will release all our codebase so that any future readers interested in this level of detail can refer to our code.
>
> > *Lines 186-190 need a better explanation for the formulations of the slow weights. More capacity and additional memory are too vague. Why formulate the fast weights with 1-D states and weights as 2-D states?*
>
> We are not sure we understand which “formulations of the slow weights” the reviewer is referring to.
> However, the reviewer is right to point out the clarity issue on Lines 189-190:
> “In contrast to regular RNNs, all these fast weight variants maintain the standard one-dimensional states and the two-dimensional fast weight states which both serve as additional memory.”
> Our original intention was to simply note that we can categorize our models as a memory augmented recurrent NN with a one-dimensional RECURRENT state (an RNN state vector) and a 2-dimensional memory (a matrix of fast weights). But it is true that this sentence does not provide much information to the readers, and citations (e.g. Graves’ Neural Turing Machine) were also missing. As this was also pointed out by another reviewer, we will edit this sentence in the revised version. We thank the reviewer for pointing this out.
>
> > *Since some of these models are not limited by the token span as much as vanilla transformers, why not try the models that can support it much longer sequence spans?*
>
> Yes, that is actually what we did in our language modelling experiment: the last two rows of Table 1 (+ full-context) show the performance of our models without explicitly limiting the context window. We obtain better performance than the baseline Transformer (first row) with contexts limited to a window of 256 tokens.
>
> > *Lines 216-218 like earlier lines are too vague, an intuition for why the slow network requires additional input (and also untested). It seems like the lack of strong reasoning behind these architecture modifications makes conclusions drawn from the results ambiguous.*
>
> We are not fully sure what “additional input” the reviewer is referring to, and also not sure which model is “untested” according to the reviewer.
> However, we agree with the reviewer that Lines 216-218 are a bit too dense. As this DeltaMLP model is not at the center of this work, we will remove that sentence from the main text, reformulate it, and put it in the appendix/supplemental material Sec A where we are already presenting an additional ablation study on DeltaMLP (see Table 4 in the supplemental material). We thank the reviewer for pointing this out.
>
> On the "*lack of strong reasoning*": as we already noted above, our empirical exploration can not be done at the level of hypothesis/test experimentation expected by the reviewer. However, our motivation for this exploration is clear. We start by looking at the regular linear Transformer through the lens of Fast Weight Programmers i.e. systems of two networks where one (the slow net) produces weights for the other one (the fast net). This perspective reveals that both nets (slow and fast) are simply one-layer feedforward networks. This naturally motivates practitioners to empirically investigate the potential gain of replacing these simple one-layer feedforward networks by traditionally more powerful models such as recurrent neural networks.
>
> > *The surprisingly good performance of the LSTM on the code execution task is also not explained or analyzed. The lack of analysis here is unfortunate because this deviates from what is expected and is worth understanding.*
>
> Whether this performance is surprising or not is debatable. We would have rather expected the LSTM to perform very well on this type of task which requires to maintain values of a few symbols. We also note that the previous work (Fan et al. [41]) reported their LSTM baseline to poorly perform on this task (certainly the difference to ours come from hyper-parameter tuning), which we found surprising and so did one of the reviewers of said work (see AnonReviewer2: https://openreview.net/forum?id=OCm0rwa1lx1).

---

> > ### Author Response · Authors · 2021-08-09
> > **Response to Reviewer u3vG, part 2/2**
> >
> > > *The falloff in LSTM performance at higher depth (depth 15) is also an interesting observation that should be expanded upon.*
> >
> > The reviewer is right to point this out. We indeed argued in Lines 282-285 that the possible explanation is the small size of the LSTM, but without providing any experimental evidence. To experimentally support this claim, we conducted an extra ablation study for this rebuttal:
> >
> > Hidden size | Test accuracy
> >
> > 256: 24.4 \% +- 1.1 (from Table 2)
> >
> > 1024: 24.4 \% +- 0.7
> >
> > 2048: 35.9 \% +- 13.0
> >
> > 4096: 72.2 \% +- 1.6
> >
> > We observe that the LSTM indeed requires much larger hidden size to perform well for this depth-15 problem.
> > We will report this result in the appendix of a revised version (Sec. B.3), and we thank the reviewer for bringing this to our attention.
> >
> > We hope that we managed to convince the reviewer on the nature of our work which is an empirical exploration.
> > While we do recognise the cleanness of the hypothesis/test style, we believe that our empirical exploration also can be valued by practitioners interested in the increasingly popular linear Transformers. We believe that this is reflected by the scores of two other reviewers.
> >
> > If our responses successfully resolved any of the reviewer’s concerns, we would really appreciate it if the reviewer would consider increasing the score.
> > Otherwise, we would appreciate it if the reviewer can clarify her/his remaining criticisms for rating our work as 4.
> > Thank you.
> >
> > [1'] Jozefowicz et al. An empirical exploration of recurrent network architectures. ICML 2015

---

> > > ### Comment · Reviewer_u3vG · 2021-08-17
> > > **Reply**
> > >
> > > Thank you for the responses and I am going to increase my reviewer score based on this discussion and clarifications. I had some additional followup.
> > >
> > > > Yes, that is actually what we did in our language modelling experiment: the last two rows of Table 1 (+ full-context) show the performance of our models without explicitly limiting the context window. We obtain better performance than the baseline Transformer (first row) with contexts limited to a window of 256 tokens.
> > >
> > > Is the backpropagation truncation at the same window of 256 tokens despite the forward pass having full-context? Is mini-batching and sampling performed differently in the full-context configuration given that there would need to be continuity between previous input?
> > >
> > > > Regarding 42-43, we are not fully sure what contradiction the reviewer is pointing at. Could you please explain? The corresponding lines are: “In the case of existing linear Transformer variants, the slow and fast nets are simple one layer feedforward NNs. Here we augment them with recurrent connections to obtain recurrent FWPs (RFWPs)”. But we can not find any contradiction here.
> > >
> > > Without the context of the next sections of the paper, the concept that "the slow and fast nets are simple one layer feedforward NNs" for a linear transformer is confusing (at least for this reviewer during the first pass).

---

> > > > ### Author Response · Authors · 2021-08-18
> > > > **Thank you and response to additional questions**
> > > >
> > > > Thank you very much for your consideration and the updated score.
> > > >
> > > > > *Is the backpropagation truncation at the same window of 256 tokens despite the forward pass having full-context? Is mini-batching and sampling performed differently in the full-context configuration given that there would need to be continuity between previous input?*
> > > >
> > > > Yes, the backpropagation span is limited to 256 tokens even for the full-context models.
> > > > The mini-batching strategy is also the same for all models: it is done such that it preserves the continuity of text between two consecutive batches (even if the models with a limited context window do not make use of such a property), which is a common/default setting for WikiText-103.
> > > > We will add this description in the appendix of a revised version.
> > > > We thank the reviewer for pointing this out.
> > > >
> > > > > *Without the context of the next sections of the paper, the concept that "the slow and fast nets are simple one layer feedforward NNs" for a linear transformer is confusing (at least for this reviewer during the first pass).*
> > > >
> > > > Thank you for the clarification.
> > > > We will append "(reviewed in Sec. 2.2)" to the corresponding sentence to guide the readers.

---

### Official Review · Reviewer_x5eT · 2021-07-20

**Rating:** 5
**Confidence:** 4

**Summary:**

The paper extends the previous line of work on interpreting linear transformer as fast weight programmers (FWP), and introduce recurrent variants of FWP.

**Main Review:**

Originality:
- The idea is not very novel in my opinion. It is a straightforward extension of the previous work, by changing either the slow network or the fast network from a feedforward neural network to a recurrent neural network.

Clarity:
- The paper is written very nicely. When introducing the background material, it provides a unifying view of the subject. It helps me understand FWP much better and also the connection between FWP and linear transformer.


Quality:
- As the paper introduces a simple variant of the existing work, I would mainly hope that the empirical results to be strong enough to deserve a publication. However, the current results do not convince me at all. Mainly,
   1. The results are not very strong. E.g., in Table 2, LSTM seems to beat all the variants proposed.
   2. The comparisons of existing works are missing. In the wikiText dataset, it missed the results from [3], which attained better performance than any of the reported variants. For CodeExec task, it missed the results from [4], which also did better than the reported results. For atari, it missed the baselines such as [1] or [2]. Note that these references are already cited in the paper (some tasks are taken from those). So it seems the author knows these results but intentionally choose to not include these as baselines?
- Over the past several years, many of the impressive results on realistic benchmarks were demonstrated without the use of recurrence. Hence this raises a question -- when is recurrence really needed? Part of the reason we prefer non-recurrence architecture is because of its easiness for scalability, and fast training speed, as one can parallelize computation across time. I would like to see a the necessity of recurrence on some practically relevant task.

Question:
- In the paper line 235-247 the authors discussed the computational efficiency of this model. But I am a bit confused by the measurement there. The computational complexity of vanilla transformer computes the embeddings for each token in the layer in one big matrix multiplication which can be highly parallelized in modern accelerators, but the recurrence would not allow computational complexity any lower than O(N), where N is the sequence length. But somehow the number of tokens per second is roughly the same. I wonder what's the reason here?


Significance:
- The work is a straightforward extension of the previous work. The results are not very convincing. Hence I wouldn't think the work is significant at its current stage.

Summary:
- I would encourage the authors to improve the empirical results to make it more convincing. In particular, I think the author should demonstrate on a practically relevant task that recurrence is necessary.


[1] Mott et.al.: Towards interpretable reinforcement learning using attention augmented agents.

[2] Parisotto et.al.: Stabilizing Transformers for reinforcement learning.

[3] Peng et. al.: Random feature attention.

[4] Fan et.al: Addressing some limitations of Transformers with feedback memory

**Time Spent Reviewing:**

4

---

> ### Author Response · Authors · 2021-08-09
> **Response to Reviewer x5eT**
>
> We thank the reviewer for valuable feedback. We believe that we have good explanations to resolve the reviewer’s main concerns. Please find our responses below.
>
> (Originality)
>
> > *The idea is not very novel in my opinion. It is a straightforward extension of the previous work, by changing either the slow network or the fast network from a feedforward neural network to a recurrent neural network.*
>
> We fully agree with the reviewer that our models are rather straightforward extensions of the existing DeltaNet model (disregarding the actual implementation and experimental evaluation).
> However, we believe our work does provide novel contributions to the existing works, mainly from the following perspectives:
>
> ***Novel exploration on linear Transformer architectures.*** Looking at the regular linear Transformer through the lens of Fast Weight Programmers, i.e., systems of two networks where one (the slow net) produces weights for the other one (the fast net), it is clear that both slow and fast nets are simply one-layer feedforward networks. From this perspective, it naturally begs the question on potentials and limits of replacing these simple one-layer feedforward networks by traditionally more powerful models such as recurrent neural networks. While this seems straightforward, this natural question had to be investigated, but nobody has created such an extension before. We thus believe that our work provides a novel architectural exploration on increasingly popular linear Transformer models.
>
> ***(Linear) Transformers for RL.*** In addition, the application of Transformer models to reinforcement learning still remains underexplored. Only one publication [1] reports results on the classic Atari environment (but only as a side experiment), and in particular, no previous work studied linear Transformers in RL, which are conceptually nicer than the regular Transformer in auto-regressive scenarios dealing with sequences of arbitrary lengths. Our work evaluates the linear Transformers in RL for the first time (to the best of our knowledge), and also demonstrates that our new model (Recurrent DeltaNet) clearly outperforms the regular linear Transformers with experiments conducted on 20 Atari games (extra results were presented in Appendix/Supplemental material Sec. C). We believe that this work contributes to the current community effort in exploring Transformers in RL (e.g., Decision Transformers [5] and Trajectory Transformers [6] which were made public on arXiv after our NeuRIPS submission; we will cite them in a revised version).
>
> (Quality)
>
> > *1.  “The results are not very strong. E.g., in Table 2, LSTM seems to beat all the variants proposed.”*
>
> While the primary goal of Table 2 is to compare the performance of different Transformer variants under the same hyper-parameter configuration, we agree with the reviewer that the low performance of the proposed models compared to the LSTM baseline shows a need for further investigation. In fact, the choice of hyper-parameters used for all Transformers presented in Table 2 is simply based on Fan et al. [2].
> We did conduct an ablation study on the CodeExec task which was included in the Appendix/Supplemental Material Table 6 (see the supplemental material from Line 668 for the full discussion). And we observed that with increased depths (8 layers), the proposed models do become more competitive compared to the LSTM.
>
> > *2 “The comparisons of existing works are missing. … So it seems the author knows these results but intentionally choose to not include these as baselines?”*
>
> The baseline numbers from [1], [2], [3], and [4] are not presented in our work because they are not comparable and/or such a comparison is irrelevant.
> We argue as follows.
>
> ***For [1] Mott et al. and [2] Parisotto et al. (RL/Atari).*** First of all, in the RL/Atari experiments, our goal is not to compare to existing baselines/SoTAs (which would require some industry level resource), but to compare different architectures under comparable conditions.
> That having said, we do report one comparison with the state-of-the-art performance as a side note (see Line 319-321), but we also clearly state that this comparison is not fair as they train a multi-task model, while we train expert models.
>
> More generally, as the reviewer certainly knows, comparison of RL results across different settings/softwares is difficult (see e.g. Andrychowicz et al. [7]). In particular, neither [1] nor [2] has released their code (both are industry papers). For [1], there is even a paper which attempted to reproduce the results with limited success [8]. Regarding [2], their setting is a (compute intense) multi-task Atari-57 which is not comparable to ours as we train separate game-specific experts.
>
> ***For [3] Peng et al. (LM/Wikitext-103).*** Our experimental setting is based on that of Schlag et al's [9] which is publicly available. While both Peng et al. [3] and Schlag et al. [9] use a similar “small” model size setting for language modelling on WikiText-103, the exact settings are not identical. Most importantly, the backpropagation span for training is 512 tokens for Peng et al’s, while Schlag et al use 256 tokens which is largely in favor of Peng’s setting but at the cost of higher compute requirement. A direct comparison is thus not fair. In addition, Schlag et al,’s code is publicly available, while Peng et al.’s is not (again an industry paper). Since our goal was not to achieve a SoTA on this special WikiText-103 setting but to compare and evaluate different Transformer variants under the same condition, both settings/baselines are equally good for the purpose of our study (so the publicly available one is more convenient).
>
> ***For [4] Fan et al. (CodeExec).*** The direct comparison with Fan et al. [4] was also not possible because the synthetic dataset was generated by Fan et al. and to the best of our knowledge, their codebase was not publicly available at the time of submission (yet again an industry paper). In consequence, we had to write our own code for generating the dataset by following the description of Fan et al. Our goal was not to achieve a state of the art on that synthetic dataset, but to study different “linear Transformer variants” under the same configurations.
>
> We really hope that these responses resolve the reviewer’s main concerns: 1 and 2.
> We take this opportunity to stress that, we will release our ENTIRE codebase (which was already provided in the supplemental material).
>
> > *“Over the past several years, … when is recurrence really needed? … I would like to see a the necessity of recurrence on some practically relevant task.”*
>
> “When is recurrence really needed?” That is a difficult open question, and we do not have a general answer to that.
> In this work, the question we asked is rather “when does recurrence help?” by empirically exploring various task domains, which is a more practical and recurrent research topic, e.g., see Tran et al. [10]. Here we had an opportunity to investigate this question in the context of the recently proposed linear Transformers which could easily be extended to be recurrent (but was not investigated previously). We find that on the code execution task and on some of the Atari environments (see Figure 8 in the Appendix which compares Recurrent DeltaNet and DeltaNet) the models with recurrence tend to outperform their feedforward counterparts.
> It is difficult to agree on what are “practically relevant" tasks, but synthetic algorithmic tasks for supervised learning and game environments in RL are common tasks for the evaluation of academic models, which is the scope of our study.
>
> (Question)
>
> > *“In the paper line 235-247 the authors discussed the computational efficiency of this model … But somehow the number of tokens per second is roughly the same. I wonder what's the reason here?”*
>
> The explanation for that is given in Lines 239 to 240: “All models are implemented using a custom CUDA kernel except the baseline Transformer for which we use regular PyTorch code.” The custom CUDA kernel is faster than the plain PyTorch code. Strictly speaking, it is not fair to compare clock-times as they are implementation-dependent. However, we preferred to provide the clock time information as-is in our experiments, so that the readers can get at least some practical sense of computational time requirement for various models we are presenting.
>
> We really hope our responses above resolve many concerns that the reviewer had raised.
> If such is the case, we would really appreciate it a lot if the reviewer could consider increasing the score.
> Thank you.
>
> [1] Mott et al. Towards interpretable reinforcement learning using attention augmented agents.
>
> [2] Parisotto et al. Stabilizing Transformers for reinforcement learning.
>
> [3] Peng et al. Random feature attention.
>
> [4] Fan et al. Addressing some limitations of Transformers with feedback memory
>
> [5] Chen et al: Decision Transformer: Reinforcement Learning via Sequence Modeling https://arxiv.org/abs/2106.01345
>
> [6] Janner et al. Reinforcement Learning as One Big Sequence Modeling Problem https://arxiv.org/abs/2106.02039
>
> [7] Andrychowicz et al. What Matters for On-Policy Deep Actor-Critic Methods? A Large-Scale Study. ICLR 2021.
> https://arxiv.org/abs/2006.05990
>
> [8] Lovering et al. Reproducing “Towards Interpretable ReinforcementLearning Using Attention Augmented Agents” https://openreview.net/forum?id=BJgtDa9GaH
>
> [9] Schlag et al. Linear Transformers Are Secretly Fast Weight Programmers. https://arxiv.org/abs/2102.11174
> (Reference [18] in our submission)
>
> [10] Tran et al. The importance of being recurrent for modeling hierarchical structure. EMNLP 2018. ([48] in our submission)

---

> > ### Author Response · Authors · 2021-08-31
> > **Friendly reminder**
> >
> > This is just a friendly reminder about the NeurIPS rebuttal deadline.
> > Please let us know if you have any remaining questions. Thank you!

---

### Decision · Program_Chairs · 2021-09-27

**Decision:**

Accept (Poster)

**Comment:**

This paper uses the connection between linear attention transforms and fast weight programmers to introduce a novel transformer architecture, recurrent fast weight programmers, which includes recurrence in the fast weights.

The authors demonstrate competitive performance on three distinct benchmarks and provide interesting conceptual insights into the functioning of their new architecture.



The discussion focussed on the role of the recurrence (“Why is it needed?”) and novel baselines / ablations. In particular, the authors provided new experimental evidence showing that the recurrence indeed helps, even when compared to a feedforward NN with the same number of parameters.

The clarifications and new evidence provided made one reviewer increase their score, while all other reviews remained unchanged.



Given the positive reviews, the discussion and the overall contributions of the paper, I recommend this paper to be accepted.